# Normalization Layers Are All That Sharpness-Aware Minimization Needs

**Maximilian Müller**
University of Tübingen
and Tübingen AI Center
maximilian.mueller@wsii.uni-tuebingen.de

**Tiffany Vlaar**
McGill University and
Mila - Quebec AI Institute
tiffany.vlaar@mila.quebec

**David Rolnick**
McGill University and
Mila - Quebec AI Institute
drolnick@cs.mcgill.ca

**Matthias Hein**
University of Tübingen
and Tübingen AI Center
matthias.hein@uni-tuebingen.de

## Abstract

Sharpness-aware minimization (SAM) was proposed to reduce sharpness of minima and has been shown to enhance generalization performance in various settings. In this work we show that perturbing only the affine normalization parameters (typically comprising 0.1% of the total parameters) in the adversarial step of SAM can outperform perturbing all of the parameters. This finding generalizes to different SAM variants and both ResNet (Batch Normalization) and Vision Transformer (Layer Normalization) architectures. We consider alternative sparse perturbation approaches and find that these do not achieve similar performance enhancement at such extreme sparsity levels, showing that this behaviour is unique to the normalization layers. Although our findings reaffirm the effectiveness of SAM in improving generalization performance, they cast doubt on whether this is solely caused by reduced sharpness.

## 1 Introduction

Numerous works have been dedicated to studying the potential connection between flatness of minima and generalization performance of deep neural networks [34, 18, 21, 45, 4]. Several aspects of training are thought to affect sharpness, but how these interact with each other remains an ongoing area of research. Recently, sharpness-aware minimization (SAM) has become a popular approach to actively try to find minima with low sharpness using a min-max type algorithm [23]. SAM was found to be remarkably effective in enhancing generalization performance for various settings [23, 14, 7, 1, 33].

Several variants of SAM have been proposed, focusing both on enhanced performance [37, 35] and reduced computational cost [11, 20]. In particular, with the original aim of making SAM more efficient Mi et al. [42] propose a 'sparse' SAM approach, which applies SAM only to a select number of parameters. Although they do not actually succeed in reducing wall-clock time, they make the surprising observation that using 50% (in some settings even up to 95%) sparse perturbations can maintain or even enhance performance compared to applying SAM to all parameters. They thus hypothesize that "complete perturbation on all parameters will result in suboptimal minima".

Similar to the effect of SAM [23, 14], normalization layers are thought to reduce sharpness [41, 46]. Frankle et al. [24] found for ResNets that the trainable affine parameters of the normalization layers

Code is provided at https://github.com/mueller-mp/SAM-ON.

37th Conference on Neural Information Processing Systems (NeurIPS 2023).

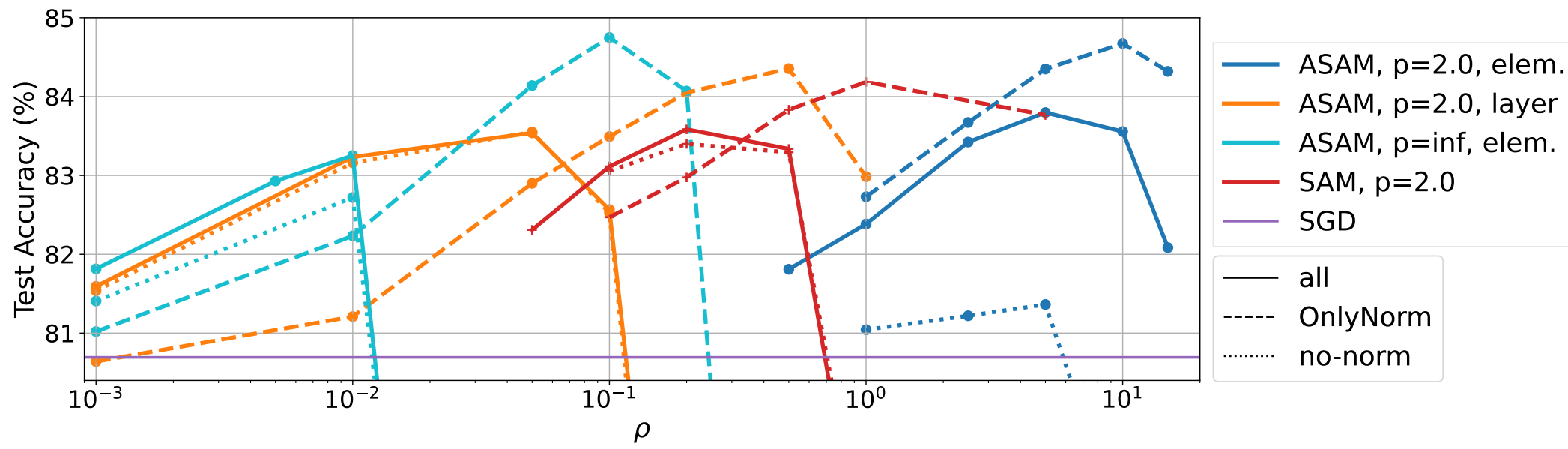

Figure 1: **The interplay of normalization layers with SAM:** Perturbing *only* normalization layers (OnlyNorm, dashed) improves generalization performance, while omitting them in the perturbation (no-norm, dotted) can harm training. WideResNet-28-10 trained with different SAM-variants on CIFAR-100. Best seen in color.

have remarkable representation capacity in their own right, whereas disabling them can reduce performance. Inspired by this we focus on the interplay between SAM and normalization layers and show that for various settings perturbing exclusively the normalization layers of a network (often less than 0.1% of the total parameters) outperforms perturbing all parameters. We find that this can not be solely attributed to possible benefits of sparse perturbation approaches and highlight the unique role played by the normalization layer affine parameters. As our main contributions we show that:

- Applying SAM only to the normalization layers of a network (*SAM-ON*, short for *SAM-OnlyNorm*) enhances performance on CIFAR data compared to applying SAM to the full network (*SAM-all*) and also performs competitively on ImageNet. We corroborate the remarkable generalization performance of *SAM-ON* for ResNet and Vision Transformer architectures, across different SAM variants, and for different batch sizes. *(Section 4)*
- Alternative sparse perturbation approaches do not result in similar performance as *SAM-ON*, especially not at the extreme sparsity levels of our method. *(Section 5.1)*
- Similar to *SAM-all*, *SAM-ON* yields non-trivial adversarial robustness (Section 4.2). It also reduces the feature-rank, but the sharpness-reducing qualities of *SAM-all* are not fully preserved (Section 5.2).

## 2 Related Work

**Normalization layers.** Batch Normalization (BatchNorm) [31] and Layer Normalization (LayerNorm) [6] form an essential component of most convolutional [25, 29] and Transformer [51, 19] architectures, respectively. Across various works these normalization layers were shown to accelerate and stabilize training, reducing sensitivity to initialization and learning rate [13, 5, 60, 36]. But despite their widespread adoption and illustrated effectiveness, a conclusive explanation for their success is still elusive. The original motivation for BatchNorm as reducing internal covariance shift [31] has been disputed [46]. The hypothesis that normalization layers enhance smoothness is supported through both empirical and theoretical analyses [46, 10, 41], though also not completely undisputed [57]. Unlike LayerNorm, BatchNorm is sensitive to the choice of batch size [38, 49]. Ghost BatchNorm, where BatchNorm statistics from disjoint subsets of the batch are used, is found to regularize and generally enhance generalization [28, 49] even though it reduces smoothness [17].

**Affine parameters.** There are relatively few papers that study the role of the trainable affine parameters of the normalization layers. Frankle et al. [24] were able to obtain surprisingly high performance on vision data by only training the BatchNorm layers, illustrating the expressive power of the affine parameters, which potentially achieve this by sparsifying activations. For BatchNorm in ResNets, disabling the affine parameters was shown empirically to reduce generalization performance [24], but for LayerNorm in Transformers to not affect or even improve performance [56]. For few-shot transfer tasks disabling the BatchNorm affine parameters during pretraining was found to enhance performance [58]. Further, many other aspects of training will have a non-trivial effect, e.g. applying weight decay to the BatchNorm affine parameters was found to increase performance for ResNets but harm performance in other settings [49].

**Sharpness-aware minimization.** SAM was developed to try to actively seek out minima with low sharpness [23]. Training with SAM may lead to increased sparsity of active neurons [14] and models which are more compressible [44]. SAM has been shown to be effective in enhancing generalization performance in various settings, but also increases the computational overhead compared to base optimizers [23, 14, 7]. Hence there have been several approaches to try to reduce the computational cost of SAM, such as ESAM which utilizes sharpness-sensitive data selection and perturbs only a randomly selected fraction of parameters [20]. Related work shows that "only employing 20% of the batch to compute the gradients for the ascent step, ... [can] result in equivalent performance" [11] and that only applying SAM to part of the parameters (SSAM) using e.g. a Fisher-information mask can lead to enhanced performance [42]. A common variant of SAM utilizes $m$-sharpness [23], which uses subbatches of size $m$ and benefits performance [20, 2, 43] though nuances in its implementation vary [8]. Andriushchenko and Flammarion [2] argue that its success is not unique to settings with BatchNorm and hence cannot be attributed to the Ghost BatchNorm effect. Andriushchenko et al. [3] further show that SAM leads to low-rank features. We discuss different SAM variants in Section 3.2.

# 3 Background: SAM and Normalization Layers

In this paper, we focus on the interplay between two popular aspects of neural network training, both of which we recapitulate here: In Sec. 3.1 we provide an overview of normalization layers, in particular BatchNorm and LayerNorm, and in Sec. 3.2 of Sharpness-Aware Minimization variants.

## 3.1 BatchNorm and LayerNorm

Modern neural network architectures typically incorporate normalization layers. In this work we will focus on Batch Normalization (BatchNorm) [31] and Layer Normalization (LayerNorm) [6], which are an essential building block of most convolutional [25, 29] and transformer [51, 19] architectures, respectively. Normalization layers transform an input $\mathbf{x}$ according to

$$N(\mathbf{x}) = \gamma \times \frac{\mathbf{x} - \mu}{\sigma} + \beta \tag{1}$$

where $\mu$ and $\sigma^2$ are the mean and variance, which are computed over the batch dimension in the case of BatchNorm, or over the embedding dimension, in the case of LayerNorm. BatchNorm is therefore sensitive to the choice of batch size [38, 49]. For BatchNorm, $\mu$ and $\sigma$ are computed from the current batch-statistics during training, and running estimates are used at test time. In our experiments, we focus on the trainable parameters $\gamma$ and $\beta$, which perform an affine transformation of the normalized input.

## 3.2 SAM and its variants

We recapitulate SAM [23], ASAM [37] and Fisher-SAM [35] with their respective perturbation models. To this end, we consider a neural network $f_{\mathbf{w}} : \mathbb{R}^d \longrightarrow \mathbb{R}^k$ which is parameterized by a vector $\mathbf{w}$ as our model. The training dataset $S^{train} = \{(\mathbf{x}_1, \mathbf{y}_1), ...(\mathbf{x}_n, \mathbf{y}_n)\}$ consists of input-output pairs which are drawn from the data distribution $D$ and we write the loss function as $l : \mathbb{R}^k \times \mathbb{R}^k \longrightarrow \mathbb{R}_+$. The goal is to learn a model $f_{\mathbf{w}}$ with good generalization performance, i.e. low expected loss $L_D(\mathbf{w}) = \mathbb{E}_{(\mathbf{x},\mathbf{y})\sim D}[l(\mathbf{y}, f_{\mathbf{w}}(\mathbf{x}))]$ on the distribution $D$. The training loss can be written as $L(\mathbf{w}) = \frac{1}{n}\sum_{i=1}^{n} l(\mathbf{y}_i, f_{\mathbf{w}}(\mathbf{x}_i))$. Conventional SGD-like optimization methods minimize (a regularized version of) $L$ by stochastic gradient descent. SAM aims at additionally minimizing the worst-case sharpness of the training loss in a neighborhood defined by an $\ell_p$ ball around $\mathbf{w}$, i.e. $\max_{||\epsilon||_p \leq \rho} L(\mathbf{w} + \epsilon) - L(\mathbf{w})$. This leads to the overall objective

$$\min_{\mathbf{w}} \max_{||\epsilon||_p \leq \rho} L(\mathbf{w} + \epsilon). \tag{2}$$

In practice, SAM uses $p = 2$ and approximates the inner maximization by a single gradient step, yielding $\epsilon = \rho \nabla L(\mathbf{w})/||\nabla L(\mathbf{w})||_2$ and requiring an additional forward-backward pass compared to SGD. The gradient is then re-evaluated at the perturbed point $\mathbf{w} + \epsilon$, giving the actual weight update

$$\mathbf{w} \longleftarrow \mathbf{w} - \alpha \nabla L(\mathbf{w} + \epsilon) \tag{3}$$

with learning rate $\alpha$. Computing $\epsilon$ separately for the batch of each GPU in multi-GPU settings and then averaging the resulting perturbed gradients for the update step in Eq. (3) has been shown to

increase SAM's performance [23]. This method is called $m$-sharpness, with $m$ being the number of samples on each GPU. Since the perturbation model in Eq. (2) is not invariant with respect to a rescaling of the weights that leaves $f_{\mathbf{w}}$ invariant [18], ASAM [37], a partly scale-invariant version of SAM, was proposed, with the objective

$$\min_{\mathbf{w}} \max_{||T_w^{-1}\epsilon||_p \leq \rho} L(\mathbf{w}+\epsilon) \tag{4}$$

where $T_w$ is a normalization operator, making the perturbation adaptive to the scale of the network parameters. Kwon et al. [37] choose $T_w$ to be diagonal with entries $T_w^i = |w_i| + \eta$ for weight parameters and $T_w^i = 1$ for bias parameters, called *elementwise* normalization. $\eta$ is typically set to $0.01$. As with SAM, the inner maximization is solved by a single gradient step:

$$\epsilon_2 = \rho \frac{T_w^2 \nabla L(\mathbf{w})}{||T_w \nabla L(\mathbf{w})||_2} \text{ for } p=2, \qquad \epsilon_\infty = \rho T_w \text{sign}\big(\nabla L(\mathbf{w})\big) \text{ for } p=\infty. \tag{5}$$

We note that for $T_w$ equal to the identity matrix and $p=2$, this is equivalent to the original SAM formulation. Recently, Kim et al. [35] proposed to use a distance metric induced by the Fisher information instead of a Euclidean distance measure between parameters. The approach can also be framed as a variant of ASAM, with $T_w$ being diagonal with entries $T_w^i = 1/\sqrt{1+\eta f_i}$ and $f_i$ approximating the $i^{th}$ diagonal entry of the Fisher-matrix by the squared average batch-gradient, $f_i = (\partial_{w_i} L_{Batch}(\mathbf{w}))^2$. For our experiments, we additionally employ layerwise normalization. This is, we set the diagonal entries of $T_w^i = ||\mathbf{W}_{\text{layer}[i]}||_2$, which corresponds to a normalization with respect to the $\ell_2$-norm of a layer, similar to [39].

# 4 SAM-ON: Perturbing Only the Normalization Layers

We study the effect of applying SAM (and its variants) solely to the normalization layers of a considered model. We will refer to this approach as *SAM-ON* (SAM-OnlyNorm) throughout this paper and provide a convergence analysis for *SAM-ON* in Appendix C. We find that *SAM-ON* obtains enhanced generalization performance compared to conventional SAM (denoted as *SAM-all*) for ResNet architectures with BatchNorm (Section 4.1) and Vision Transformers with LayerNorm (Section 4.2) on CIFAR data and performs competitively on ImageNet. For comparison, we also study the reverse of *SAM-ON*, i.e. we exclude the affine normalization parameters from the adversarial SAM-step, which we shall refer to as *no-norm*.

**Training set-up.** We use SGD with momentum, weight decay, and cosine learning rate decay as our base optimizer for ResNet architectures and employ label smoothing to adopt similar settings as in the literature [37]. For Vision Transformers we employ AdamW [40] as our base optimizer on CIFAR and for ImageNet we additionally use Lion [15]. We use both basic augmentations (random cropping and flipping) and strong augmentations (basic+AutoAugment, denoted as +AA). We consider a range of SAM-variants which differ either in the perturbation model ($\ell_2$ or $\ell_\infty$) or in the definition of the normalization operator. We train models for 200 epochs, and do not employ $m$-sharpness unless indicated otherwise. Complete training details are described in Appendix A.

## 4.1 BatchNorm and ResNet

**CIFAR.** We showcase the effect of *SAM-ON*, i.e. only applying SAM to the BatchNorm parameters, for a WideResNet-28-10 (WRN-28) on CIFAR-100 in Figure 1. We observe that *SAM-ON* obtains higher accuracy than conventional SAM (*SAM-all*) for all SAM variants considered (more SAM-variants are shown in Figure 6 in the Appendix). In contrast, excluding the affine BatchNorm parameters from the adversarial step (*no-norm*) either significantly decreases performance (for elementwise-$\ell_2$ variants) or maintains similar performance as *SAM-all* (for all other variants). For variants which do not experience a performance drop for *no-norm*, the ideal SAM perturbation radius $\rho$ shifts towards larger values, indicating that the perturbation model cannot perturb the BatchNorm parameters enough when *all* parameters are used. To study if the benefits of only perturbing the normalization layers extends to other settings, we train more ResNet-like models on CIFAR-10 and CIFAR-100. For each SAM-variant and dataset, we probe a set of pre-defined $\rho$-values (shown in Table 7 in the Appendix) with a ResNet-56 (RN-56) and fix the best-performing $\rho$ for the other models to compare *SAM-ON* to *SAM-all*. We report mean accuracy and standard deviation over 3 seeds for CIFAR-100 in Table 1. On average, *SAM-ON* outperforms *SAM-all* for all considered

Table 1: ***SAM-ON* improves over *SAM-all* for BatchNorm and ResNets**: Test accuracy for ResNet-like models on CIFAR-100. Bold values mark the better performance between *SAM-ON* and *SAM-all* within a SAM-variant, and underline highlights the overall best method per model and augmentation.

| | | RN-56 [25] | | RNxT [55] | | WRN-28 [59] | |
|---|---|---|---|---|---|---|---|
| | variant | all | ON | all | ON | all | ON |
| basic aug. | SGD | $72.82^{\pm0.3}$ | | $80.16^{\pm0.3}$ | | $80.71^{\pm0.2}$ | |
| | SAM | $75.07^{\pm0.6}$ | $\mathbf{75.58}^{\pm0.4}$ | $81.79^{\pm0.4}$ | $\mathbf{82.22}^{\pm0.2}$ | $83.11^{\pm0.3}$ | $\mathbf{84.19}$$^{\pm0.2}$ |
| | el. $\ell_2$ | $75.05^{\pm0.1}$ | $\mathbf{76.25}$$^{\pm0.0}$ | $81.26^{\pm0.2}$ | $\mathbf{82.30}^{\pm0.3}$ | $82.38^{\pm0.2}$ | $\mathbf{83.67}^{\pm0.3}$ |
| | el. $\ell_2$, orig. | $75.54^{\pm0.7}$ | $\mathbf{76.07}^{\pm0.2}$ | $\mathbf{82.15}^{\pm0.3}$ | $81.90^{\pm0.4}$ | $\mathbf{83.67}^{\pm0.1}$ | $83.53^{\pm0.2}$ |
| | el. $\ell_\infty$ | $75.36^{\pm0.1}$ | $\mathbf{76.10}^{\pm0.2}$ | $81.02^{\pm0.6}$ | $\mathbf{82.38}^{\pm0.3}$ | $83.25^{\pm0.2}$ | $\mathbf{84.14}^{\pm0.2}$ |
| | Fisher | $75.01^{\pm0.4}$ | $\mathbf{75.65}^{\pm0.1}$ | $81.55^{\pm0.2}$ | $\mathbf{82.21}^{\pm0.2}$ | $83.37^{\pm0.1}$ | $\mathbf{84.01}^{\pm0.1}$ |
| | layer. $\ell_2$ | $74.63^{\pm0.1}$ | $\mathbf{76.03}^{\pm0.3}$ | $81.66^{\pm0.2}$ | $\mathbf{82.52}$$^{\pm0.2}$ | $83.23^{\pm0.2}$ | $\mathbf{84.05}^{\pm0.2}$ |
| basic aug. + AA | SGD | $75.26^{\pm0.2}$ | | $80.31^{\pm0.3}$ | | $83.62^{\pm0.1}$ | |
| | SAM | $\mathbf{76.33}^{\pm0.3}$ | $76.02^{\pm0.3}$ | $82.33^{\pm0.5}$ | $\mathbf{83.19}^{\pm0.2}$ | $85.30^{\pm0.1}$ | $\mathbf{85.42}^{\pm0.1}$ |
| | el. $\ell_2$ | $\mathbf{76.51}^{\pm0.1}$ | $76.04^{\pm0.3}$ | $82.00^{\pm0.3}$ | $\mathbf{83.20}^{\pm0.1}$ | $84.80^{\pm0.3}$ | $\mathbf{85.43}^{\pm0.3}$ |
| | el. $\ell_2$, orig. | $76.49^{\pm0.2}$ | $\mathbf{76.58}^{\pm0.4}$ | $82.78^{\pm0.1}$ | $\mathbf{82.87}^{\pm0.3}$ | $85.25^{\pm0.4}$ | $\mathbf{85.41}^{\pm0.1}$ |
| | el. $\ell_\infty$ | $74.89^{\pm0.4}$ | $\mathbf{76.19}^{\pm0.4}$ | $82.33^{\pm0.1}$ | $\mathbf{83.11}^{\pm0.2}$ | $85.28^{\pm0.1}$ | $\mathbf{85.46}^{\pm0.1}$ |
| | Fisher | $\mathbf{76.67}^{\pm0.1}$ | $76.25^{\pm0.2}$ | $82.56^{\pm0.3}$ | $\mathbf{83.28}^{\pm0.4}$ | $85.09^{\pm0.3}$ | $\mathbf{85.35}^{\pm0.1}$ |
| | layer. $\ell_2$ | $76.23^{\pm0.5}$ | $\mathbf{76.93}$$^{\pm0.4}$ | $82.61^{\pm0.3}$ | $\mathbf{83.32}$$^{\pm0.2}$ | $85.32^{\pm0.3}$ | $\mathbf{85.95}$$^{\pm0.1}$ |

SAM-variants. Of these, layerwise-$\ell_2$ achieves the highest performance for most settings. We obtain similar results for CIFAR-10 (App. Table 10) and for more network architectures (App. Table 11).

**ImageNet.** For ImageNet, we adopt the timm training script [53]. We train a ResNet-50 for 100 epochs on eight 2080-Ti GPUs with $m = 64$, leading to an overall batch-size of 512. Apart from $\rho$, all hyperparameters are shared for all SAM-variants and can be found in the Appendix in Table 8. We select the most promising *SAM-ON* variants and compare them against the established methods (SGD, SAM, ASAM elementwise $\ell_2$). The results are shown in Table 2. We observe that for layerwise $\ell_2$, the *all* variant achieves higher accuracy, whereas the *SAM-ON* models outperform their *all* counterparts for elementwise $\ell_2$ and elementwise $\ell_\infty$. All *SAM-ON* variants outperform the previously established methods (SGD, SAM, ASAM). For reference, we also show the values reported for ESAM [20] and GSAM [61], two SAM-variants we did not include in our study.

Table 2: ImageNet top-1 accuracy for a ResNet-50. ESAM [20] and GSAM [61] values are taken from the respective papers.

| SGD | SAM | ESAM | GSAM | elem. $\ell_2$ | | elem. $\ell_\infty$ | | layer $\ell_2$ | |
|---|---|---|---|---|---|---|---|---|---|
| | all | all | all | all | ON | all | ON | all | ON |
| $77.03^{\pm0.13}$ | $77.65^{\pm0.11}$ | 77.05 | 77.20 | $77.65^{\pm0.05}$ | $\mathbf{77.82}^{\pm0.14}$ | $77.45^{\pm0.04}$ | $\mathbf{77.82}^{\pm0.01}$ | $\mathbf{78.14}$$^{\pm0.05}$ | $77.87^{\pm0.07}$ |

**Varying the batchsize.** In Figure 4 (right) we report the performance of a WRN-28 on CIFAR-100 with *SAM-ON* and *SAM-all* for a range of batch-sizes and values of $m$, where $m$ is the batch-size per accelerator, as discussed in Section 3.2. Similar to the findings in [23, 2], we confirm that lower values of $m$ lead to better performance within each batch-size. Importantly, *SAM-ON* outperforms *SAM-all* for all combinations of batch-size and $m$, illustrating that Ghost BatchNorm [28, 49] (see discussion in Section 2) does not play a role in the success of *SAM-ON*.

### 4.2 LayerNorm and Vision Transformer

**CIFAR.** To study the effectiveness of *SAM-ON* beyond ResNet architectures and BatchNorm, we train ViTs from scratch on CIFAR data with AdamW as the base optimizer (Figure 2). Although ResNet architectures are known to outperform Vision Transformers when trained from scratch on small-scale datasets like CIFAR, our aim here is not to outperform state-of-the-art, but rather to study if the benefits of *SAM-ON* extend to substantially different training settings. Remarkably, we find that the same phenomena occur: The *SAM-ON* variants outperform their conventional counterparts *SAM-all* by a clear margin. For the elementwise-$\ell_2$ variants there is a strong drop in accuracy for *no-norm*, whereas for the other SAM variants the optimal perturbation radius $\rho$ shifts towards larger values. We show that this extends to a ViT-T and CIFAR-10 as well (Table 3).

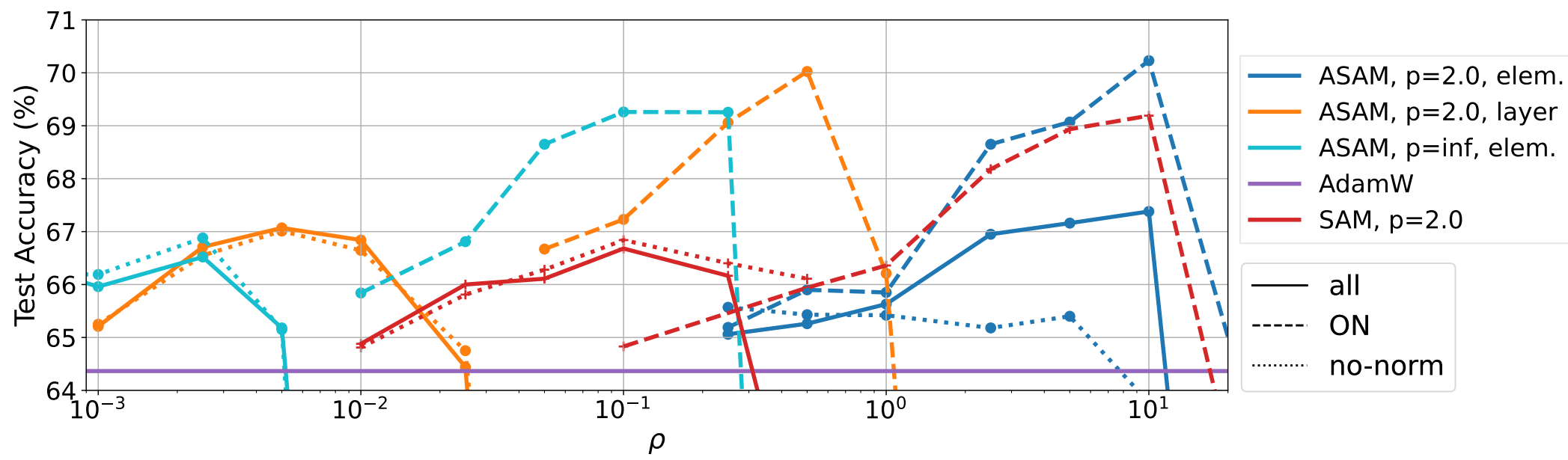

Figure 2: **The interplay of normalization layers with SAM for ViT training** : ViT-S trained with different SAM-variants on CIFAR-100. Like for ResNets, perturbing *only* normalization layers (OnlyNorm) improves generalization performance, while omitting them in the perturbation (no-norm) can harm training.

Table 3: ***SAM-ON* improves over *SAM-all* for LayerNorm and ViTs**: Shown are ViT models on CIFAR-10 and CIFAR-100. Bold values mark the better performance between *SAM-ON* and *SAM-all* within a SAM-variant, and underline highlights the overall best method per model.

| | | CIFAR-10 | | | | CIFAR-100 | | | |
|---|---|---|---|---|---|---|---|---|---|
| | | ViT-T | | ViT-S | | ViT-T | | ViT-S | |
| | variant | all | ON | all | ON | all | ON | all | ON |
| basic aug. + AA | AdamW | $89.19^{\pm 0.2}$ | | $90.34^{\pm 0.0}$ | | $63.79^{\pm 0.0}$ | | $64.37^{\pm 0.2}$ | |
| | SAM | $88.92^{\pm 0.0}$ | $\mathbf{92.22}^{\pm 0.1}$ | $90.81^{\pm 0.2}$ | $\mathbf{92.71}^{\pm 0.0}$ | $64.70^{\pm 0.4}$ | $\mathbf{69.84}^{\pm 0.2}$ | $66.58^{\pm 0.3}$ | $\mathbf{69.13}^{\pm 0.3}$ |
| | el. $\ell_2$ | $90.02^{\pm 0.2}$ | $\mathbf{92.52}^{\pm 0.3}$ | $92.40^{\pm 0.4}$ | $\mathbf{93.97}^{\pm 0.2}$ | $65.74^{\pm 0.6}$ | $\mathbf{71.09}^{\pm 0.1}$ | $66.98^{\pm 0.0}$ | $\mathbf{70.31}^{\pm 0.2}$ |
| | el. $\ell_2$, orig. | $90.22^{\pm 0.2}$ | $\mathbf{92.68}^{\pm 0.2}$ | $92.05^{\pm 0.3}$ | $\mathbf{94.01}^{\pm 0.5}$ | $65.81^{\pm 0.6}$ | $\underline{\mathbf{71.25}}^{\pm 0.3}$ | $67.23^{\pm 0.2}$ | $\underline{\mathbf{70.42}}^{\pm 0.5}$ |
| | el. $\ell_\infty$ | $89.82^{\pm 0.3}$ | $\mathbf{92.66}^{\pm 0.2}$ | $90.76^{\pm 0.6}$ | $\mathbf{93.68}^{\pm 0.2}$ | $65.11^{\pm 0.3}$ | $\mathbf{68.11}^{\pm 0.7}$ | $66.55^{\pm 0.1}$ | $\mathbf{69.19}^{\pm 0.1}$ |
| | Fisher | $89.08^{\pm 0.1}$ | $\mathbf{92.03}^{\pm 0.2}$ | $91.13^{\pm 0.2}$ | $\mathbf{92.49}^{\pm 0.1}$ | $64.70^{\pm 0.4}$ | $\mathbf{69.55}^{\pm 0.8}$ | $66.59^{\pm 0.5}$ | $\mathbf{69.30}^{\pm 0.4}$ |
| | layer. $\ell_2$ | $89.51^{\pm 0.4}$ | $\underline{\mathbf{93.08}}^{\pm 0.2}$ | $91.21^{\pm 0.1}$ | $\underline{\mathbf{94.02}}^{\pm 0.1}$ | $65.30^{\pm 0.6}$ | $\mathbf{69.66}^{\pm 0.1}$ | $67.39^{\pm 0.4}$ | $\mathbf{70.04}^{\pm 0.2}$ |

**ImageNet.** For ImageNet, we train a ViT-S/32 from scratch with batchsize 128 on a single GPU for 300 epochs. In Table 4 we evaluate *SAM-all*, *SAM-ON* and the vanilla variant for both AdamW and Lion [15] as base optimizers on both ID and OOD datasets. Since Wei et al. [52] showed that SAM-trained models show non-trivial robustness to small adversarial perturbations, we also investigate the robustness of *SAM-ON* (last rows of Table 4). Both *SAM-all* and *SAM-ON* improve strongly over the base optimizer in all setups. For AdamW, *SAM-ON* either outperforms *SAM-all* (e.g. w.r.t. adversarial robustness) or performs on par (e.g. for ID accuracy the numbers are within standard deviations). For Lion, *SAM-ON* always outperforms *SAM-all*, underlining that the diverse benefits of SAM can be achieved or even surpassed by only perturbing the normalization layers. We provide all experimental details in in Appendix A.2 and a more thorough evaluation and discussion of the adversarial robustness of *SAM-all* and *SAM-ON* in Appendix B.4.

Table 4: **SAM-ON performs well on ImageNet:** Training a ViT-S/32 from scratch.

| | | AdamW | | | Lion | | |
|---|---|---|---|---|---|---|---|
| | | vanilla | SAM-all | SAM-ON | vanilla | SAM-all | SAM-ON |
| ID | ImageNet | $66.89^{\pm 0.04}$ | $71.47^{\pm 0.12}$ | $71.37^{\pm 0.026}$ | $68.20^{\pm 0.02}$ | $71.90^{\pm 0.19}$ | $\mathbf{72.64}^{\pm 0.14}$ |
| | ImageNetV2 | $48.43^{\pm 0.48}$ | $53.61^{\pm 0.11}$ | $53.67^{\pm 0.29}$ | $50.20^{\pm 0.01}$ | $54.20^{\pm 0.27}$ | $\mathbf{55.38}^{\pm 0.09}$ |
| OOD | ImageNetR | $25.04^{\pm 0.04}$ | $31.56^{\pm 0.48}$ | $\mathbf{32.98}^{\pm 0.10}$ | $25.61^{\pm 0.04}$ | $32.17^{\pm 0.41}$ | $\mathbf{33.87}^{\pm 0.47}$ |
| | ImageNetA | $4.72^{\pm 0.15}$ | $5.21^{\pm 0.05}$ | $5.19^{\pm 0.18}$ | $5.45^{\pm 0.19}$ | $5.01^{\pm 0.22}$ | $\mathbf{5.77}^{\pm 0.21}$ |
| | ImageNetSketch | $13.68^{\pm 0.24}$ | $18.50^{\pm 0.44}$ | $\mathbf{19.35}^{\pm 0.17}$ | $14.47^{\pm 0.02}$ | $18.22^{\pm 0.34}$ | $\mathbf{20.48}^{\pm 0.12}$ |
| | ObjectNet | $11.32^{\pm 0.39}$ | $13.75^{\pm 0.12}$ | $13.55^{\pm 0.25}$ | $12.06^{\pm 0.02}$ | $13.93^{\pm 0.40}$ | $\mathbf{15.35}^{\pm 0.13}$ |
| adv. rob. | $\ell_2, \epsilon = 0.25$ | $19.67^{\pm 0.47}$ | $37.53^{\pm 0.69}$ | $\mathbf{41.16}^{\pm 0.24}$ | $22.01^{\pm 0.78}$ | $38.52^{\pm 0.66}$ | $\mathbf{43.12}^{\pm 0.97}$ |
| | $\ell_2, \epsilon = 0.50$ | $5.47^{\pm 0.18}$ | $17.71^{\pm 0.61}$ | $\mathbf{22.72}^{\pm 0.25}$ | $6.63^{\pm 0.46}$ | $19.03^{\pm 0.92}$ | $\mathbf{24.27}^{\pm 1.34}$ |
| | $\ell_\infty, \epsilon = 0.25/255$ | $33.45^{\pm 0.80}$ | $48.08^{\pm 0.14}$ | $\mathbf{49.34}^{\pm 0.08}$ | $35.31^{\pm 0.08}$ | $49.57^{\pm 0.60}$ | $\mathbf{51.37}^{\pm 0.99}$ |
| | $\ell_\infty, \epsilon = 0.5/255$ | $14.98^{\pm 0.18}$ | $29.68^{\pm 0.09}$ | $\mathbf{32.46}^{\pm 0.15}$ | $15.86^{\pm 0.13}$ | $31.68^{\pm 0.62}$ | $\mathbf{34.23}^{\pm 1.73}$ |

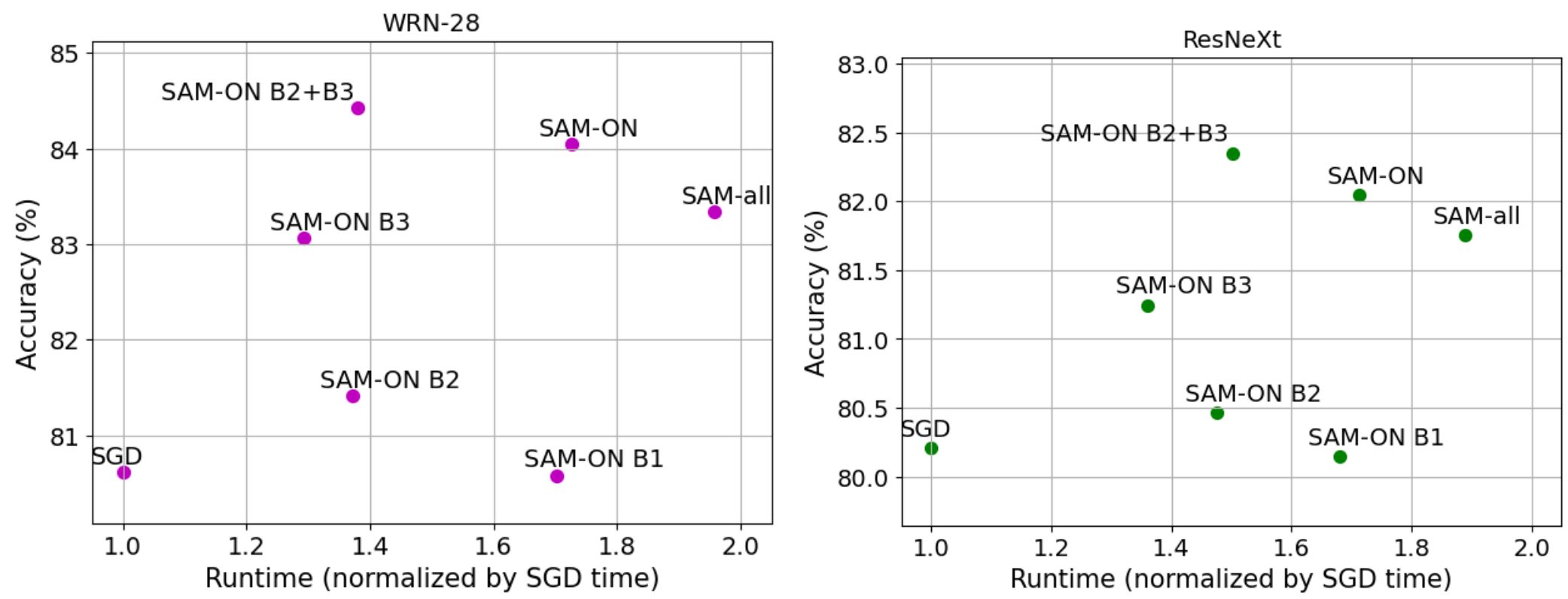

Figure 3: **Computational gains of SAM-ON over SAM-all:** Test accuracy vs. normalized wall-clock runtime for SAM and different variations of SAM-ON for a WRN-28 (left) and a ResNeXt (right) on CIFAR-100. Only perturbing selected normalization parameters (e.g. those from block 3, or those from block 2 *and* 3) can lead to further computational gains. Reported values are averaged over three random seeds.

### 4.3 Computational savings

In Figure 3 we report the wall-clock time of training a WRN-28 (left) and a ResNeXt (right) with batchsize 128 on a single A100 with PyTorch. Since the normalization parameters at the earlier layers of the network require a gradient, potentially a full backpropagation pass has to be computed for the ascent-step of *SAM-ON*, even though only a tiny fraction of all parameters is perturbed. However, as discussed in [12] for a related setting, the gradients of the intermediate (no-norm) layers do not need to be stored or used for updating. This leads to computational gains of *SAM-ON* over *SAM-all* as reported in Figure 3.

In contrast, although future development of hardware for sparse operation may allow for acceleration, Mi et al. [42] report that their sparse perturbation approach does not at present lead to reduced wall-clock time. Their approach also suffers from additional computational cost associated with selecting the mask, and hence is outshined by *SAM-ON* both in computational cost and generalization performance (as discussed in Section 5.1).

We also show results for perturbing only the normalization layers of selected blocks (Block 1-3) of the network, and interestingly the main benefits of *SAM-ON* seem to arise from the later normalization layers, allowing for further computational savings. When perturbing only the normalization layers from the last block (B3), the biggest computational gains can be achieved (reducing the additional cost of SAM by more than 50%), without much loss in test accuracy. When perturbing the normalization layers from Block 2 and 3 (B2+B3), the test accuracy even slightly improves over *SAM-ON* for both models, while the runtime is still significantly lower. We have not investigated this variant of *SAM-ON* thoroughly, i.e. in combination with other ASAM perturbation models and more network architectures, but think that this is an interesting research direction for future work.

## 5 Towards Understanding SAM-ON

To gain a better understanding of *SAM-ON*, we study different hypotheses for the method's success. First, we investigate the role of sparsity by comparing *SAM-ON* to different sparsified perturbation approaches (Section 5.1), concluding that sparsity alone is not enough to explain its success. Then, we highlight that *SAM-ON* might in fact find *sharper* minima, while generalizing better than *SAM-all* (Section 5.2). We also show that *SAM-ON* - similar to *SAM-all* - reduces the feature-rank compared to vanilla optimizers (Section 5.2). Further, we showcase that depending on the perturbation method, *SAM-ON* can induce a significant shift in the distribution of the normalization parameters (Section 5.3) and relate this to the *no-norm* results from Section 4. Unless stated otherwise, we use a WRN-28 with BatchNorm and the setting described in Section 4 for the ablation studies. Further ablation studies are presented in Appendix B.

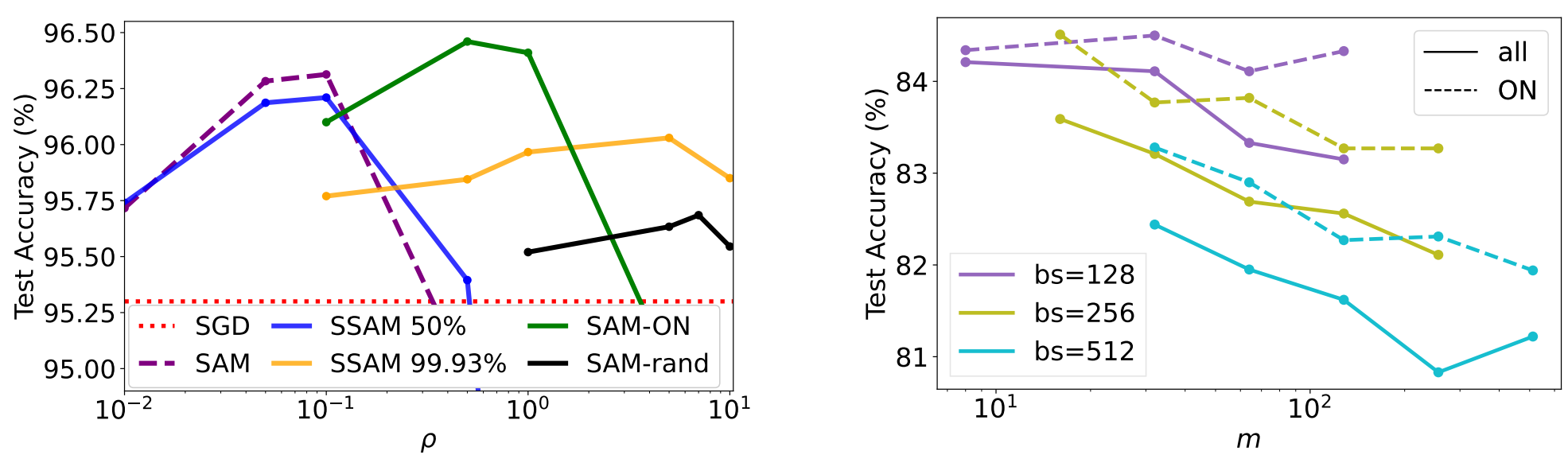

Figure 4: **Left:** *SAM-ON* outperforms *SSAM-F* [42] (with different sparsity levels) and random mask *SAM-rand* (same sparsity level 99.93% as *SAM-ON*) sparse perturbation approaches on CIFAR-10 for ResNet-18. **Right:** *SAM-ON* improves over SAM for various batch-sizes (bs) and values of $m$, where smaller values of $m$ tend to improve performance. WRN-28, CIFAR-100.

### 5.1 The effect of sparsified perturbations

Mi et al. [42] propose a sparsified SAM (*SSAM*) approach which only considers part of the parameters for the SAM perturbation step. They find that using 50% perturbation sparsity *SSAM* can outperform *SAM-all*, and still perform competitively with up to 99% sparsity in certain settings. This raises the question whether the enhanced performance of *SAM-ON* over *SAM-all* is simply due to perturbing fewer parameters. We therefore compare *SAM-ON* to other sparse perturbation approaches.

In order to determine the parameters which should be perturbed, one solution Mi et al. [42] propose is to compute a binary mask via an approximation of the Fisher matrix. We call this approach *SSAM-F* to avoid confusion with Fisher-SAM introduced in Section 3.2. Mi et al. [42] also propose a dynamic mask sparse perturbation approach (*SSAM-D*), but do not clearly favour either method. Since they consider *SSAM-F* "relatively more stable but a bit time-consuming, while *[SSAM-D]* is more efficient", we will for fair comparison focus on *SSAM-F* here and provide results for *SSAM-D* in Appendix B.2. According to Mi et al. [42] neither approach improves wall-clock time in practice, while we find that *SAM-ON* does (see Section 4.3).

We provide a comparison between *SAM-ON*, *SSAM-F*, and a random sparsity mask of the same sparsity level as *SAM-ON* for *a)* ResNet-18 on CIFAR-10 (main setting considered in [42]) in Figure 4 (left) and *b)* WRN-28 on CIFAR-100 in Table 5. We find that although *SSAM-F* can indeed perform on par or even outperform *SAM-all* at the medium to high sparsity levels recommended in [42] it is less successful than *SAM-ON*. Moreover, for the sparsity levels of *SAM-ON*, which are above 99.9% in both settings, a random mask performs poorly. These results suggest that sparsity is not the sole cause for *SAM-ON*'s success.

Table 5: **SAM-ON outperforms other sparse perturbation approaches:** Although *SSAM-F* [42] with different sparsity levels can outperform *SAM-all* on CIFAR-100 with WRN-28, it is less effective than *SAM-ON*, especially when probed at very high sparsity levels.

| | SAM | SAM-ON | Random Mask | SSAM-F | |
|---|---|---|---|---|---|
| Sparsity | 0% | 99.95% | 99.95% | 50% | 99.95% |
| Test Accuracy (%) | 83.11±0.3 | **84.19**±0.2 | 80.97±0.2 | 83.94±0.1 | 83.14±0.1 |

### 5.2 Sharpness and feature-rank of SAM-ON

The improved generalization performance of SAM-trained models is often attributed to finding flatter minima [27, 34] – indeed this was the initial motivation behind SAM in [23]. Andriushchenko and Flammarion [2] however cast doubt on this explanation, and argue that the benefits of SAM might stem from a favorable implicit bias induced by the method. A recent study furthermore found that sharpness often does not correlate well with a model's generalization performance [4]. In this section we therefore compare the sharpness of *SAM-all* to *SAM-ON* models (following the setup in [4], more details provided in Appendix B.10).

In Table 6, we report logit-normalized worst-case adaptive $\ell_\infty$ $m$-sharpness, the sharpness definition which achieved the best correlation with generalization error for CIFAR data in the large-scale study in [4]. We investigate a WRN-28 on CIFAR-100 trained with SGD, SAM, and ASAM-$\ell_\infty$ both with their respective *all* and *ON* variants.

Worst-case sharpness is measured with 20 steps of AutoPGD [16], a hyperparameter-free method designed for accurate estimation of adversarial robustness. Instead of the input space, which is optimized in adversarial robustness, the method is adapted to the weight space for sharpness computation. In addition to the 20-step AutoPGD-sharpness, we also report 1-step sharpness, i.e. the change in loss obtained when performing only a single gradient step, like in the perturbation step of SAM. It is to note that except for the logit-normalization, the sharpness definition reported in Table 6 corresponds exactly to the perturbation model that ASAM elementwise $\ell_\infty$ uses, and hence the 1-step sharpness reported should be fairly close the the objective that ASAM elementwise $\ell_\infty$ actually minimizes during training. For both sharpness definitions, we pick $\rho$ such that we get sharpness values similar to those that lead to good correlation with generalization in [4]. This is, for 1-step sharpness we have to pick $\rho$ larger than for the 20-step sharpness to get to similar loss changes.

We observe that the *SAM-ON* models obtain the best generalization performance, while having significantly higher sharpness than their *SAM-all* counterparts, and sometimes even higher than that of the baseline SGD-trained model. This supports the claims of [2] that although *SAM-all* might in fact find flatter minima, we should be careful in attributing this as the sole reason for its improved generalization performance. In Appendix B.10 we report more sharpness metrics and find that *SAM-ON* is sharper than *SAM-all* for most metrics, although there exist exceptions.

A recent study by Andriushchenko et al. [3] further showed that *SAM-all* leads to features of lower rank compared to vanilla optimizers. Following their setup we measure the feature rank for a WRN-28 and report our findings in Table 6 (final row). We find that *SAM-ON* also leads to features of lower rank compared to SGD, but observe no significant change compared to *SAM-all*.

Table 6: ***SAM-ON* models are sharper, yet generalize better.** Shown is logit-normalized $\ell_\infty$ $m$-sharpness from [4], averaged over three models per method for a WRN-28 on CIFAR-100, and the feature rank like investigated in [3] (last row).

| | | SGD | SAM | | ASAM-el.-$l_\infty$ | |
|---|---|---|---|---|---|---|
| | | | all | ON | all | ON |
| Test Accuracy (%) | | $80.71^{\pm 0.2}$ | $83.11^{\pm 0.3}$ | $\mathbf{84.19}^{\pm 0.2}$ | $83.25^{\pm 0.2}$ | $\mathbf{84.14}^{\pm 0.2}$ |
| $\ell_\infty$-sharpness | 20 steps, $\rho = 0.003$ | $0.071^{\pm 0.000}$ | $\mathbf{0.048}^{\pm 0.001}$ | $0.090^{\pm 0.005}$ | $\mathbf{0.048}^{\pm 0.001}$ | $0.078^{\pm 0.004}$ |
| | 20 steps, $\rho = 0.005$ | $0.201^{\pm 0.001}$ | $\mathbf{0.139}^{\pm 0.004}$ | $0.296^{\pm 0.018}$ | $\mathbf{0.124}^{\pm 0.002}$ | $0.283^{\pm 0.011}$ |
| | 20 steps, $\rho = 0.007$ | $0.433^{\pm 0.002}$ | $\mathbf{0.309}^{\pm 0.011}$ | $0.585^{\pm 0.018}$ | $\mathbf{0.255}^{\pm 0.005}$ | $0.580^{\pm 0.020}$ |
| | 1 step, $\rho = 0.007$ | $0.117^{\pm 0.002}$ | $\mathbf{0.098}^{\pm 0.001}$ | $0.170^{\pm 0.007}$ | $\mathbf{0.095}^{\pm 0.002}$ | $0.142^{\pm 0.011}$ |
| | 1 step, $\rho = 0.01$ | $0.204^{\pm 0.005}$ | $\mathbf{0.183}^{\pm 0.002}$ | $0.315^{\pm 0.010}$ | $\mathbf{0.170}^{\pm 0.003}$ | $0.271^{\pm 0.019}$ |
| | 1 step, $\rho = 0.03$ | $0.809^{\pm 0.003}$ | $\mathbf{0.769}^{\pm 0.017}$ | $0.843^{\pm 0.007}$ | $\mathbf{0.724}^{\pm 0.005}$ | $0.834^{\pm 0.012}$ |
| Feature Rank | | $4004^{\pm 12}$ | $3798^{\pm 17}$ | **3728** $^{\pm 11}$ | **3613** $^{\pm 47}$ | $3725^{\pm 11}$ |

### 5.3 SAM-ON can change the affine parameter values

In Figure 5 we show the distribution of the scaling parameter $\gamma$ and the shift parameter $\beta$ (as defined in Eq. 1) for a WRN-28 trained with *SAM-ON* and *SAM-all*. While there are only minor changes in the distribution of $\beta$, there is a clear shift in the distribution of $\gamma$ towards larger values for *SAM-ON*. In Figure 9 in Appendix B.6 we study more SAM-variants and rediscover a pattern from Section 4.1: The distribution shift for $\gamma$ only seems to appear for those variants, where the optimal $\rho$ of *SAM-ON* shifts towards larger values compared to *SAM-all* (i.e. *not* for ASAM elementwise-$\ell_2$). We thus hypothesize that the perturbation model of ASAM elementwise-$\ell_2$ implicitly focuses mostly on perturbing the normalization layers already, which is why excluding them (*no-norm*) leads to a drastic performance decrease, whereas *SAM-ON* minimally changes the $\gamma$-distribution. In contrast, the other SAM-variants (for which the optimal $\rho$-value changes) may not perturb the normalization layers *enough* in *SAM-all* for them to be effective, which is why *no-norm* has little effect, while *SAM-ON* leads to a distinctive shift of the $\gamma$-distribution.

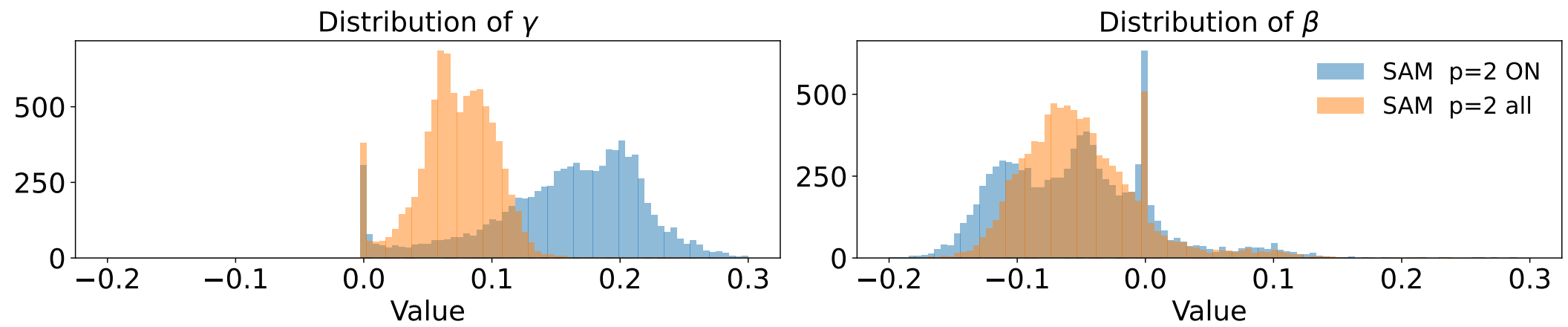

Figure 5: *SAM-ON* changes the weight distribution of the normalization layer weights of a WRN-28 towards larger values. More SAM-variants are shown in Figure 9 in the appendix.

## 6 Discussion and Conclusion

In recent years the method of sharpness-aware minimization (SAM) [23] has risen to prominence due to its demonstrated effectiveness and the community's long-standing interest in flat minima. In this work we show that only applying SAM to the normalization layers (*SAM-ON*) – typically less than $0.1\%$ of the total parameters – can significantly enhance performance compared to regular SAM (*SAM-all*). We show results on CIFAR and ImageNet for ResNet and Vision Transformers with BatchNorm and Layernorm, respectively. Although the use of sparsified perturbations was recently shown to benefit generalization [42], we show that the success of *SAM-ON* cannot be attributed to sparsity alone: targeting the normalization layers clearly improves over other masked sparsity approaches.

We find that while *SAM-ON* outperforms *SAM-all* in almost all settings, the optimal SAM variant to use varies. We do not see a consistent benefit of using reparameterization-invariant perturbation models compared to variants with fewer invariances. In particular, we find that layerwise $\ell_2$ in combination with *SAM-ON* reaches the highest accuracy for many settings.

While *SAM-ON* improves generalization, it loses some of SAM's sharpness-reducing qualities. Although perhaps surprising given SAM's original motivation, this finding relates naturally to the literature. [4] find that sharpness does not always correlate well with generalization performance. Further, [2] question if sharpness is the sole driving factor behind SAM's success in enhancing generalization. We lend support to this question, showing that SAM-like methods can generalize better, without significant sharpness reduction.

In summary, we demonstrate benefits of SAM beyond reducing sharpness and highlight the special role played by the normalization layers. More investigation into the interplay of SAM and other aspects of training are needed to fully understand where the methods gains come from.

**Limitations.** Similar to the main inspirations for this work [23, 24, 42] we focus on vision data. We provide a simple experiment in the language domain in Appendix B.5 indicating that the efficacy of *SAM-ON* might be preserved for language tasks, but more extensive benchmarking, as done by [7] is required. Further, more work is required to try to leverage the perturbation sparsity for reduced computational cost beyond the gains obtained in this work, and to fully explore the benefits of perturbing subsets of the normalization layers building on the findings in Section 4.3.

### Acknowledgements

We acknowledge support from the Deutsche Forschungsgemeinschaft (DFG, German Research Foundation) under Germany's Excellence Strategy (EXC number 2064/1, Project number 390727645), as well as from the Carl Zeiss Foundation in the project "Certification and Foundations of Safe Machine Learning Systems in Healthcare". We also thank the European Laboratory for Learning and Intelligent Systems (ELLIS) for supporting Maximilian Müller. We are grateful for support from the Canada CIFAR AI Chairs Program and US National Science Foundation award tel:1910864. In addition, we acknowledge material support from NVIDIA and Intel in the form of computational resources and are grateful for technical support from the Mila IDT team in maintaining the Mila Compute Cluster.

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

# Appendix

This appendix is structured as follows:

- In Appendix A we provide more training details. In particular, we report the hyperparameters used for the CIFAR experiments in A.1 and for the ImageNet experiments in A.2. In A.3 we provide more details and a formal definition of the SAM-variants used throughout this paper.
- In Appendix B we show additional experimental results for: CIFAR in B.1, ImageNet in B.3, and a machine translation task in B.5. In B.2 we provide additional ablation studies for sparse perturbation *SSAM* approaches and in B.4 we extend the discussion on adversarial robustness. To gain a better understanding of SAM-ON, we further investigate: the weight distribution shift induced by *SAM-ON* (B.6), the effect of SAM when fixing the normalization parameters during training (B.7), SAM's performance when only training the normalization layers (B.8), and ablations on weight decay and dropout (B.9). Finally, we provide an extended discussion on the sharpness evaluation and more ablations in B.10.
- In Appendix C we provide a convergence analysis for *SAM-ON*.

# A Training Details

## A.1 CIFAR training details

For our CIFAR experiments, we consider a range of SAM-variants which differ either in the norm ($p \in \{2, \infty\}$) or in the definition of the normalization operator. We use SGD, the original SAM with no normalization and $p = 2$, Fisher-SAM and the following ASAM-variants: elementwise-$\ell_\infty$, layerwise-$\ell_2$, and elementwise-$\ell_2$. For the ViT-experiments, we use AdamW instead of SGD. For each of the ASAM-variants, we normalize both bias and weight parameters and set $\eta = 0$. Additionally, we employ the original ASAM-algorithm, where the bias parameters are not normalized and $\eta = 0.01$. We train all models on a single GPU for 200 epochs, and m-sharpness is not employed (unless indicated otherwise). For ResNets, we follow [37] and adopt a learning rate of 0.1, momentum of 0.9, weight decay of $0.0005$ and use label smoothing with a factor of 0.1. We use both basic augmentations (random cropping and flipping) and strong augmentations (basic+AutoAugment). For ViTs we use AdamW with learning rate $0.0001$, batchsize 64 and only strong augmentations, the other settings remain unchanged. The ResNet results were computed on 2080ti-GPUs and the ViT results on A100s. The values of $\rho$ we considered for each method can be found in Table 7. The ResNet-networks we considered for the CIFAR-experiments in the main paper are ResNet56 (RN56) [25], ResNeXt-29-32x4d (RNxT) [55], and WideResNet-28-10 (WRN) [59]. We adopted the ViTs to CIFAR by setting the image-size to 32 and patch-size to 4.

Table 7: Search-space for $\rho$. The values used for the the experiments in Tables 1,3 and 10 are marked in bold.

| | | CIFAR-10 RN | CIFAR-100 RN | CIFAR-10/100 ViT |
|---|---|---|---|---|
| SAM | all | 0.05, **0.1**, 0.25 | 0.05, **0.1**, 0.5, 1. | 0.025, 0.05, **0.1**, 0.25, 0.5 |
| SAM | ON | 0.1, **0.5**, 1 | 0.1, 0.5, **1.**, 5. | 1., 2.5, 5., **10.**, 25 |
| el. $l_2$ | all | 0.5, 1, **2**, 3, 5 | 0.5, **1**, 2.5, 5., 10. | 0.5, 1., **2.5**, 5, 10 |
| el. $l_2$ | ON | 0.5, 1, 2, **3**, 5 | 0.5, 1., **2.5**, 5., 10. | 1., 2.5, 5.,**10.**, 25 |
| el. $l_2$, orig. | all | 0.1, **0.5**, 1, 5, 10 | 0.5, **1**, 2.5, 5 | 0.5, 1., **2.5**, 5, 10 |
| el. $l_2$, orig. | ON | 0.1, **0.5**, 1, 5, 10 | 0.5, **1.**, 2.5, 5 | 1., 2.5, 5.,**10.**, 25 |
| el. $l_\infty$ | all | 0.001, **0.005**, 0.01, 0.05 | 0.001, 0.005, **0.01**, 0.05 | 0.0005, 0.001, **0.0025**, 0.005, 0.01 |
| el. $l_\infty$ | ON | 0.01, **0.025**, 0.05, 0.1 | 0.01, **0.05**, 0.1, 0.5 | 0.025, 0.05, **0.1**, 0.25, 0.5 |
| layer $l_2$ | all | 0.005, 0.01, **0.025**, 0.05, 0.1 | 0.001, **0.01**, 0.05, 0.1 | 0.001, 0.0025, **0.005**, 0.01, 0.025 |
| layer $l_2$ | ON | 0.05, 0.1, **0.25**, 0.5, 1 | 0.1, **0.2**, 0.5, 1. | 0.05, 0.1, 0.25, **0.5**, 1. |
| Fisher | all | 0.05, **0.1**, 0.5, 1,5 | 0.05, **0.1**, 0.5, 1 | 0.05,5 **0.1**, 0.5, 1,5 |
| Fisher | ON | 0.1, **0.5**,1 ,5 , 10 | 0.1, 0.5,**1** ,5 , 10 | 0.1, 0.5,1 ,5 , **10** |

## A.2 ImageNet training details

Table 8 shows the hyperparameters for all variants used for ImageNet training. For the ResNet-50 with SGD, SAM and elementwise-$\ell_2$ we used the hyperparameters from [23] and [37]. For the layerwise $\ell_2$ and elementwise-$\ell_\infty$ we tried two $\rho$-values per configuration and report the results of the better one (named $\rho$ (reported) in the table). $\rho$ (discarded) refers to the $\rho$ value we probed, but found to perform worse than the other one. For the ViT-S (additional fine-tuning experiments in Appendix B.3), we tried at least three values of $\rho$ per SAM-configuration and reported the best one.

Table 8: Hyperparameters for training on ImageNet. Top: ResNet-50 from scratch, center: ViT-S from scratch, bottom: finetuning the ViT-S.

| param | SGD | SAM | elem. $\ell_2$ | | ResNet-50 | elem. $\ell_\infty$ | | layer $\ell_2$ | |
|---|---|---|---|---|---|---|---|---|---|
| | all | all | all | onlyNorm | | all | onlyNorm | all | onlyNorm |
| train epochs | | | | | 90 | | | | |
| warm-up epochs | | | | | 3 | | | | |
| cool-down epochs | | | | | 10 | | | | |
| batch-size | | | | | 512 | | | | |
| augmentation | | | | | inception-style | | | | |
| lr | | | | | 0.2 | | | | |
| lr decay | | | | | Cosine | | | | |
| weight decay | | | | | 0.0001 | | | | |
| $\rho$ (reported) | | 0.05 | 1 | 1 | | 0.001 | 0.005 | 0.005 | 0.05 |
| $\rho$ (discarded) | | | | | | 0.01 | 0.05 | 0.05 | 0.5 |
| Input Resolution | | | | | $224 \times 224$ | | | | |
| m | | | | | 64 | | | | |
| GPU Type | | | | | 8×2080-ti | | | | |

| param | AdamW | AdamW+SAM | | ViT-S scratch | Lion | Lion+SAM | |
|---|---|---|---|---|---|---|---|
| | all | all | onlyNorm | | all | all | onlyNorm |
| train epochs | | | | 300 | | | |
| warm-up epochs | | | | 10 | | | |
| cool-down epochs | | | | 0 | | | |
| batch-size | | | | 128 | | | |
| augmentation | | | | inception-style | | | |
| lr | | | | 0.001 | | | |
| lr decay | | | | Cosine | | | |
| weight decay | | | | 0.1 | | | |
| $\rho$ (reported) | – | 1 | 15 | | – | 1 | 10 |
| $\rho$ (discarded) | – | 0.05,0.1,0.5,2 | 10,20 | | – | 0.5,2 | 5,20 |
| Input Resolution | | | | $224 \times 224$ | | | |
| m | | | | 128 | | | |
| GPU Type | | | | 1×A100 | | | |

| param | SGD | SAM | | elem. $\ell_2$ | | ViT-S FT | elem. $\ell_\infty$ | | layer $\ell_2$ | |
|---|---|---|---|---|---|---|---|---|---|---|
| | all | all | onlyNorm | all | onlyNorm | | all | onlyNorm | all | onlyNorm |
| train epochs | | | | | | 9 | | | | |
| warm-up epochs | | | | | | 1 | | | | |
| cool-down epochs | | | | | | 0 | | | | |
| batch-size | | | | | | 896 | | | | |
| augmentation | | | | | | inception-style | | | | |
| lr | | | | | | 0.017 | | | | |
| lr decay | | | | | | Cosine | | | | |
| weight decay | | | | | | 0.0001 | | | | |
| $\rho$ (reported) | – | 0.01 | 0.1 | 0.1 | 1 | | $10^{-4}$ | $10^{-2}$ | $10^{-3}$ | $10^{-3}$ |
| $\rho$ (discarded) | | 0.1 | 0.01 | 0.01 | 0.1 | | $10^{-3}$ | $10^{-3}$ | $10^{-2}$ | $10^{-2}$ |
| $\rho$ (discarded) | | 0.001 | 1. | 1. | 10 | | $10^{-5}$ | $10^{-1}$ | $10^{-4}$ | $10^{-1}$ |
| Input Resolution | | | | | | $224 \times 224$ | | | | |
| m | | | | | | 128 | | | | |
| GPU Type | | | | | | 7×A100 | | | | |

### A.3 SAM variants

Here, we provide a more comprehensive overview of the SAM-variants used throughout the experiments. To this end, we first recall the definition of the (A)SAM-perturbation (Eq. (5) in the main paper):

$$\epsilon_2 = \rho \frac{T_w^2 \nabla L(\mathbf{w})}{||T_w \nabla L(\mathbf{w})||_2} \text{ for } p=2, \qquad \epsilon_\infty = \rho T_w \text{sign}\big(\nabla L(\mathbf{w})\big) \text{ for } p=\infty.$$

with the normalization operator $T_w^i$, which is diagonal for all variants. We note that *SAM-ON* can be formally defined as using the conventional (A)SAM-algorithm but setting all entries $T_w^i = 0$ if $w_i$ is not a normalization parameter. This leads to a change of the perturbation $\epsilon$ according to Eq. (5). Importantly, the magnitude of $\epsilon$ is still $\rho$, since both the nominator and the denominator of Eq. (5) change. We provide an overview over all (A)SAM-variants and their respective perturbation models in Table 9.

Table 9: The definition of $T_w^i$ for the considered SAM-variants.

| variant | | $T_w^i$ | $p$ | $\eta$ |
|---|---|---|---|---|
| SAM | all | $1$ | 2 | 0 |
| | ON | $\begin{cases} 1 & \text{if } w_i \text{ is a normalization parameter} \\ 0 & \text{else} \end{cases}$ | 2 | 0 |
| el. $\ell_2$ | all | $\lvert w_i \rvert$ | 2 | 0 |
| | ON | $\begin{cases} \lvert w_i \rvert & \text{if } w_i \text{ is a normalization parameter} \\ 0 & \text{else} \end{cases}$ | 2 | 0 |
| el. $\ell_2$, orig. | all | $\begin{cases} \lvert w_i \rvert + \eta & \text{if } w_i \text{ is a weight parameter} \\ 1+\eta & \text{if } w_i \text{ is a bias parameter} \end{cases}$ | 2 | 0.01 |
| | ON | $\begin{cases} \lvert w_i \rvert + \eta & \text{if } w_i \text{ is a normalization weight} \\ 1+\eta & \text{if } w_i \text{ is a normalization bias} \\ 0 & \text{else} \end{cases}$ | 2 | 0.01 |
| el. $\ell_\infty$ | all | $\lvert w_i \rvert$ | $\infty$ | 0 |
| | ON | $\begin{cases} \lvert w_i \rvert & \text{if } w_i \text{ is a normalization parameter} \\ 0 & \text{else} \end{cases}$ | $\infty$ | 0 |
| layer $\ell_2$ | all | $\lvert\lvert \mathbf{W}_{\text{layer[i]}} \rvert\rvert_2$ | 2 | 0 |
| | ON | $\begin{cases} \lvert\lvert \mathbf{W}_{\text{layer[i]}} \rvert\rvert_2 & \text{if } w_i \text{ is a normalization parameter} \\ 0 & \text{else} \end{cases}$ | 2 | 0 |
| Fisher | all | $\left(1+\eta\left(\partial_{w_i} L_{Batch}(\mathbf{w})\right)^2\right)^{-0.5}$ | 2 | 1 |
| | ON | $\begin{cases} \left(1+\eta\left(\partial_{w_i} L_{Batch}(\mathbf{w})\right)^2\right)^{-0.5} & \text{if } w_i \text{ is a normalization parameter} \\ 0 & \text{else} \end{cases}$ | 2 | 1 |

## B Further Experimental Results

### B.1 SAM-ON on CIFAR

We omitted the results for ResNet-like models on CIFAR-10 in the main paper. Those are thus reported in Table 10. Due to the already very high accuracies, the differences between *SAM-ON* and *SAM-all* are smaller, yet on average *SAM-ON* is still clearly the better method. We further plot all considered SAM-variants for different values of $\rho$ in Figure 6 for a WRN-28 and in Figure 7 for a ViT-S on CIFAR-100. We show results for various VGG-models [47] and DenseNet-100 [30] for CIFAR-10/100 in Table 11 and observe that *SAM-ON* consistently improves over *SAM-all*.

### B.2 Additional ablation studies for sparse SAM

In this section we provide additional ablation studies for sparsified perturbation approaches as discussed in Section 5.1. Mi et al. [42] proposed two sparsified SAM (*SSAM*) approaches: Fisher SSAM (*SSAM-F*) and Dynamic SSAM (*SSAM-D*). As an extension to Figure 4 for ResNet-18 on CIFAR-10 data in the main paper we provide an accompanying Figure 8 which includes error bars

Table 10: ***SAM-ON* improves over *SAM-all* for BatchNorm and ResNets on CIFAR-10**: Test accuracy for ResNet-like models on CIFAR-10. Bold values mark the better performance between *SAM-ON* and *SAM-all* within a SAM-variant, and underline highlights the overall best method per model and augmentation

| | SAM variant | RN-56 | | RNxT | | WRN-28 | |
|---|---|---|---|---|---|---|---|
| | | all | onlyNorm | all | onlyNorm | all | onlyNorm |
| basic aug. | SGD | $94.28^{\pm0.2}$ | | $95.37^{\pm0.1}$ | | $96.20^{\pm0.1}$ | |
| | SAM | $94.94^{\pm0.1}$ | $\mathbf{95.18}^{\pm0.1}$ | $96.35^{\pm0.2}$ | $\mathbf{96.48}^{\pm0.1}$ | $97.08^{\pm0.1}$ | $\mathbf{97.10}^{\pm0.0}$ |
| | elem. $\ell_2$ | $\mathbf{94.96}^{\pm0.1}$ | $94.94^{\pm0.2}$ | $96.41^{\pm0.1}$ | $\mathbf{96.53}^{\pm0.1}$ | $96.98^{\pm0.2}$ | $\mathbf{97.06}^{\pm0.0}$ |
| | elem. $\ell_2$, orig. | $95.14^{\pm0.1}$ | $\underline{\mathbf{95.21}}^{\pm0.1}$ | $96.40^{\pm0.1}$ | $\mathbf{96.41}^{\pm0.1}$ | $\mathbf{97.10}^{\pm0.1}$ | $97.07^{\pm0.1}$ |
| | elem. $\ell_\infty$ | $94.93^{\pm0.1}$ | $\mathbf{94.96}^{\pm0.0}$ | $96.06^{\pm0.2}$ | $\mathbf{96.22}^{\pm0.1}$ | $96.95^{\pm0.2}$ | $\mathbf{97.00}^{\pm0.1}$ |
| | Fisher | $95.01^{\pm0.1}$ | $\mathbf{95.03}^{\pm0.1}$ | $96.31^{\pm0.0}$ | $\underline{\mathbf{96.55}}^{\pm0.0}$ | $96.95^{\pm0.0}$ | $\underline{\mathbf{97.13}}^{\pm0.1}$ |
| | layer. $\ell_2$ | $94.95^{\pm0.2}$ | $\mathbf{95.07}^{\pm0.1}$ | $96.07^{\pm0.3}$ | $\mathbf{96.46}^{\pm0.1}$ | $\mathbf{97.02}^{\pm0.0}$ | $96.96^{\pm0.1}$ |
| basic aug. + AA | SGD | $94.70^{\pm0.1}$ | | $96.19^{\pm0.2}$ | | $97.01^{\pm0.0}$ | |
| | SAM | $95.25^{\pm0.1}$ | $\mathbf{95.40}^{\pm0.1}$ | $96.98^{\pm0.1}$ | $\mathbf{97.22}^{\pm0.3}$ | $97.57^{\pm0.1}$ | $\mathbf{97.58}^{\pm0.0}$ |
| | elem. $\ell_2$ | $\mathbf{95.12}^{\pm0.0}$ | $94.82^{\pm0.2}$ | $97.01^{\pm0.0}$ | $\mathbf{97.21}^{\pm0.1}$ | $97.61^{\pm0.0}$ | $\underline{\mathbf{97.69}}^{\pm0.0}$ |
| | elem. $\ell_2$, orig. | $95.39^{\pm0.1}$ | $\underline{\mathbf{95.60}}^{\pm0.1}$ | $97.24^{\pm0.0}$ | $\underline{\mathbf{97.33}}^{\pm0.1}$ | $\mathbf{97.60}^{\pm0.0}$ | $97.56^{\pm0.0}$ |
| | elem. $\ell_\infty$ | $95.12^{\pm0.1}$ | $\mathbf{95.48}^{\pm0.3}$ | $96.70^{\pm0.2}$ | $\mathbf{96.91}^{\pm0.2}$ | $97.52^{\pm0.1}$ | $\mathbf{97.62}^{\pm0.1}$ |
| | Fisher | $95.19^{\pm0.0}$ | $\mathbf{95.38}^{\pm0.1}$ | $96.77^{\pm0.0}$ | $\mathbf{97.24}^{\pm0.1}$ | $97.53^{\pm0.0}$ | $\mathbf{97.65}^{\pm0.1}$ |
| | layer. $\ell_2$ | $\mathbf{95.43}^{\pm0.3}$ | $95.28^{\pm0.1}$ | $96.80^{\pm0.1}$ | $\mathbf{96.88}^{\pm0.1}$ | $\mathbf{97.60}^{\pm0.0}$ | $97.48^{\pm0.1}$ |

Table 11: ***SAM-ON* improves over *SAM-all* for BatchNorm and more ResNet models**: Bold values mark the better performance between *SAM-ON* and *SAM-all* within a SAM-variant, and underline highlights the overall best method per model and augmentation

| | SAM variant | VGG-13 | | VGG-16 | | VGG-19 | | DenseNet-100 | |
|---|---|---|---|---|---|---|---|---|---|
| | | all | onlyNorm | all | onlyNorm | all | onlyNorm | all | onlyNorm |
| CIFAR-100 | SGD | $75.44^{\pm0.2}$ | | $74.43^{\pm0.4}$ | | $73.40^{\pm0.2}$ | | $77.00^{\pm0.2}$ | |
| | SAM | $76.74^{\pm0.2}$ | $\mathbf{77.57}^{\pm0.1}$ | $75.81^{\pm0.2}$ | $\mathbf{76.86}^{\pm0.1}$ | $74.08^{\pm0.6}$ | $\underline{\mathbf{75.60}}^{\pm0.1}$ | $79.42^{\pm0.6}$ | $\mathbf{79.90}^{\pm0.3}$ |
| | elem. $\ell_2$ | $76.65^{\pm0.1}$ | $\mathbf{77.49}^{\pm0.1}$ | $75.95^{\pm0.2}$ | $\mathbf{76.45}^{\pm0.2}$ | $74.72^{\pm0.2}$ | $\mathbf{75.12}^{\pm0.1}$ | $78.90^{\pm0.2}$ | $\mathbf{79.83}^{\pm0.3}$ |
| | elem. $\ell_2, \eta = 0.01$ | $77.27^{\pm0.2}$ | $\mathbf{77.37}^{\pm0.2}$ | $76.65^{\pm0.1}$ | $\mathbf{76.66}^{\pm0.3}$ | $75.00^{\pm0.5}$ | $\mathbf{75.44}^{\pm0.2}$ | $79.94^{\pm0.4}$ | $\mathbf{80.14}^{\pm0.1}$ |
| | elem. $\ell_\infty$ | $76.82^{\pm0.3}$ | $\mathbf{77.62}^{\pm0.2}$ | $75.43^{\pm0.4}$ | $\mathbf{76.68}^{\pm0.1}$ | $72.74^{\pm0.2}$ | $\mathbf{74.50}^{\pm0.4}$ | $79.47^{\pm0.3}$ | $\mathbf{79.64}^{\pm0.2}$ |
| | Fisher, $\eta = 1.$ | $76.76^{\pm0.2}$ | $\mathbf{77.68}^{\pm0.4}$ | $75.85^{\pm0.2}$ | $\mathbf{76.99}^{\pm0.1}$ | $74.03^{\pm0.2}$ | $\mathbf{74.96}^{\pm0.3}$ | $79.68^{\pm0.2}$ | $\underline{\mathbf{80.38}}^{\pm0.3}$ |
| | layer. $\ell_2$ | $76.76^{\pm0.2}$ | $\underline{\mathbf{77.91}}^{\pm0.2}$ | $75.99^{\pm0.2}$ | $\underline{\mathbf{77.12}}^{\pm0.2}$ | $74.65^{\pm0.5}$ | $\mathbf{75.28}^{\pm0.2}$ | $78.25^{\pm0.2}$ | $\mathbf{79.86}^{\pm0.3}$ |
| CIFAR-10 | SGD | $94.29^{\pm0.0}$ | | $93.85^{\pm0.3}$ | | $93.82^{\pm0.0}$ | | $94.51^{\pm0.1}$ | |
| | SAM | $94.88^{\pm0.1}$ | $\underline{\mathbf{95.19}}^{\pm0.2}$ | $94.96^{\pm0.0}$ | $\mathbf{95.02}^{\pm0.1}$ | $94.58^{\pm0.1}$ | $\mathbf{94.81}^{\pm0.2}$ | $95.84^{\pm0.2}$ | $\mathbf{95.89}^{\pm0.0}$ |
| | elem. $\ell_2$ | $94.97^{\pm0.1}$ | $\mathbf{95.08}^{\pm0.0}$ | $95.01^{\pm0.1}$ | $\mathbf{95.02}^{\pm0.1}$ | $94.68^{\pm0.0}$ | $\underline{\mathbf{94.99}}^{\pm0.1}$ | $95.76^{\pm0.2}$ | $\mathbf{95.86}^{\pm0.2}$ |
| | elem. $\ell_2, \eta = 0.01$ | $94.95^{\pm0.0}$ | $\mathbf{95.13}^{\pm0.1}$ | $94.87^{\pm0.1}$ | $\underline{\mathbf{95.12}}^{\pm0.1}$ | $94.66^{\pm0.1}$ | $\mathbf{94.87}^{\pm0.2}$ | $\mathbf{95.92}^{\pm0.3}$ | $95.85^{\pm0.1}$ |
| | elem. $\ell_\infty$ | $94.96^{\pm0.1}$ | $\mathbf{95.06}^{\pm0.0}$ | $94.74^{\pm0.2}$ | $\mathbf{94.91}^{\pm0.0}$ | $94.68^{\pm0.1}$ | $\mathbf{94.73}^{\pm0.1}$ | $95.56^{\pm0.2}$ | $\mathbf{95.91}^{\pm0.1}$ |
| | Fisher, $\eta = 1.$ | $95.07^{\pm0.0}$ | $\mathbf{95.17}^{\pm0.0}$ | $94.77^{\pm0.0}$ | $\mathbf{95.10}^{\pm0.2}$ | $94.55^{\pm0.0}$ | $\mathbf{94.91}^{\pm0.1}$ | $95.65^{\pm0.1}$ | $\underline{\mathbf{96.00}}^{\pm0.1}$ |
| | layer. $\ell_2$ | $94.78^{\pm0.1}$ | $\mathbf{95.09}^{\pm0.1}$ | $94.54^{\pm0.1}$ | $\mathbf{95.08}^{\pm0.1}$ | $66.21^{\pm48.7}$ | $\mathbf{94.96}^{\pm0.1}$ | $95.48^{\pm0.2}$ | $\mathbf{95.82}^{\pm0.1}$ |

and comparisons with the dynamic sparse perturbation approach (*SSAM-D*) [42]. We also provide additional results for *SSAM-D* for a WideResNet-28 on CIFAR-100 data in Table 12. We found optimal performance for *SSAM* for 50% sparsity and $\rho = 0.1$ on CIFAR-10 and $\rho = 0.2$ on CIFAR-100 (as also observed in [42] for slightly different training settings). We find that although both *SSAM* approaches can perform on par or even outperform regular *SAM*, they are less effective than our *SAM-ON* approach. The generalization gap increases even further when considering the same high sparsity levels as for *SAM-ON*.

Table 12: Although *SSAM-F* and *SSAM-D* [42] with different sparsity levels can outperform *SAM-all* on CIFAR-100 with WRN-28, they are less effective than *SAM-ON*.

| | SAM | SAM-ON | SAM-rand | SSAM-F | | SSAM-D | |
|---|---|---|---|---|---|---|---|
| Sparsity | 0% | 99.95% | 99.95% | 50% | 99.95% | 50% | 99.95% |
| Accuracy | $83.11_{\pm0.3}$ | $\mathbf{84.19}_{\pm0.2}$ | $80.97_{\pm0.2}$ | $83.94_{\pm0.1}$ | $83.14_{\pm0.1}$ | $83.53_{\pm0.1}$ | $81.01_{\pm0.1}$ |

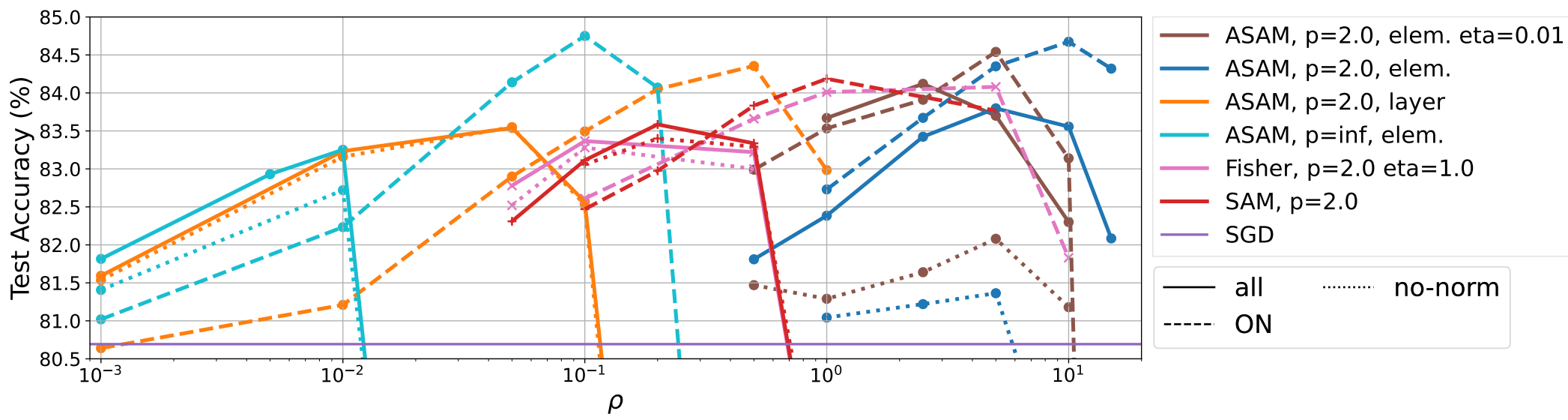

Figure 6: All considered SAM-variants and their *SAM-ON* counterpart for a WRN-28 on CIFAR-100.

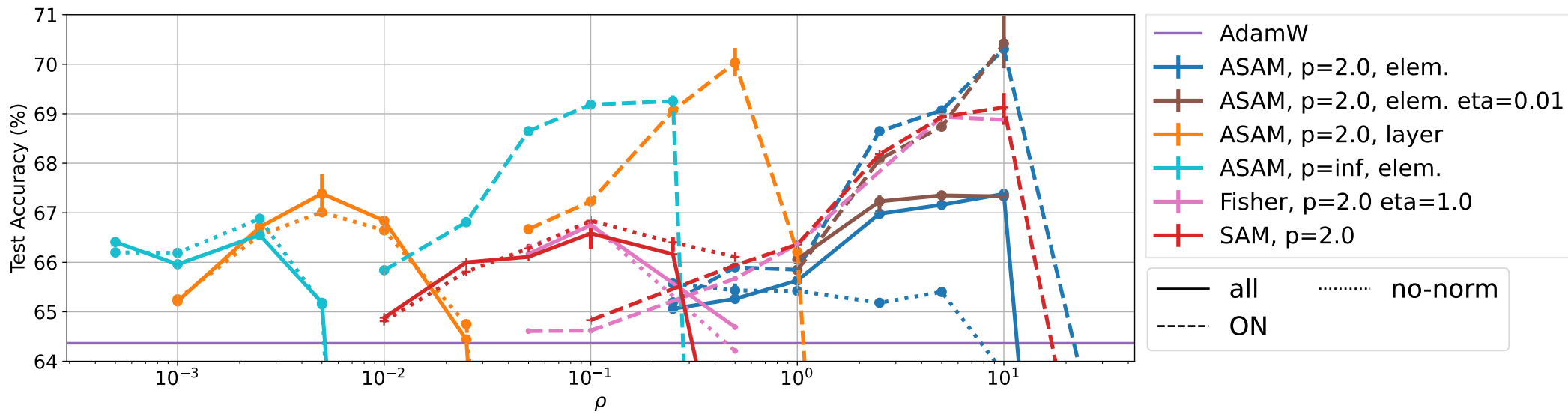

Figure 7: All considered SAM-variants and their *SAM-ON* counterpart for a ViT-S on CIFAR-100.

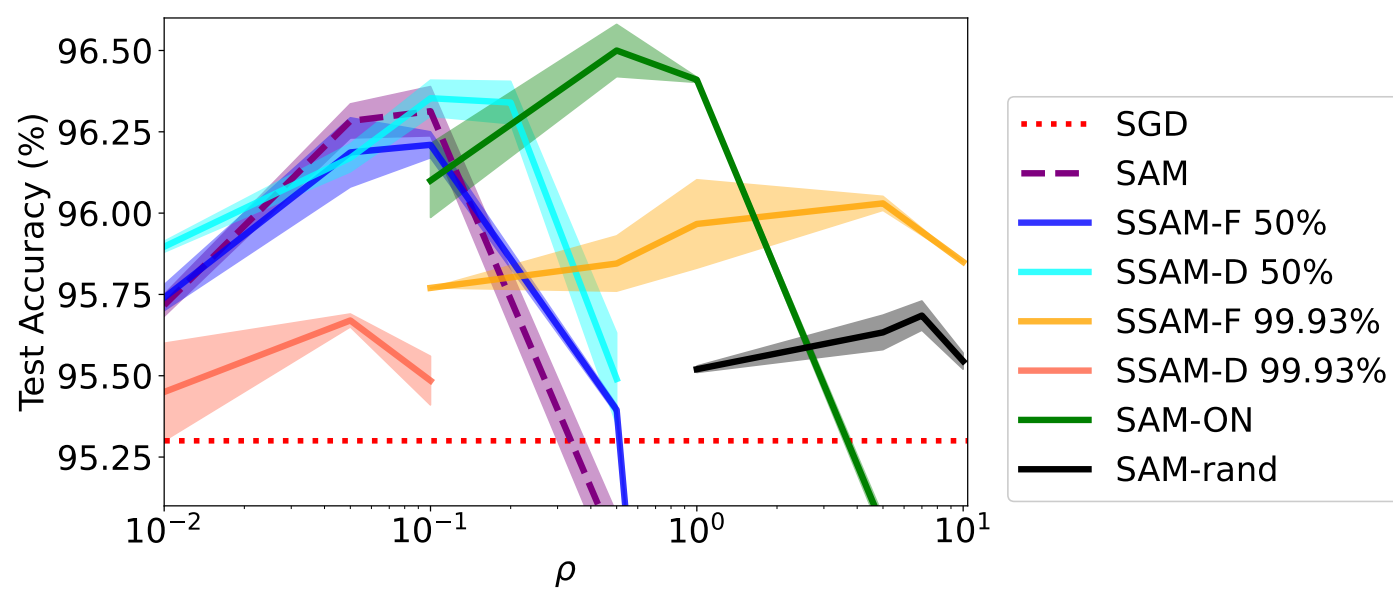

Figure 8: *SAM-ON* outperforms *SSAM-F* and *SSAM-D* [42] (with different sparsity levels) and random mask *SAM-rand* (same sparsity level 99.93% as *SAM-ON*) sparse perturbation approaches on CIFAR-10 for ResNet-18.

### B.3 Finetuning from ImageNet-21k

Since ViTs are commonly trained on large-scale datasets and then fine-tuned, we investigate this scenario for *SAM-ON*. In particular, we consider a ViT-S pretrained on ImageNet-21k from [48]. We fine-tune for 9 epochs with SGD for a range of SAM-variants with their respective *SAM-ON* counterpart. For each setup, we probe three values of $\rho$ and report the best result in Table 13. We find in this setting that *SAM-ON* performs on par with *SAM-all* although there are small differences across SAM variants: for layerwise-$\ell_2$ *SAM-ON* performs slightly worse, whereas for all other variants *SAM-ON* performs equally well or slightly better than *SAM-all*. In all cases *SAM-ON* outperforms plain SGD.

Table 13: Results for ImageNet-1k fine-tuning of a ViT-S-224 from a ImageNet-21k model.

| SGD | SAM | | ASAM elem. $\ell_2$ | | ASAM layer. $\ell_2$ | | ASAM elem. $\ell_\infty$ | |
|---|---|---|---|---|---|---|---|---|
| | all | ON | all | ON | all | ON | all | ON |
| 81.62 | **81.75** | **81.75** | 81.73 | **81.75** | **81.79** | 81.75 | **81.84** | **81.84** |

### B.4 Adversarial robustness

Here, we provide additional results and extend the discussion on adversarial robustness from Section 4.2. In a study by Wei et al. [52] SAM-trained models showed non-trivial robustness to small adversarial perturbations [50]. Since there are several works highlighting the role of normalization layers for adversarial robustness [9, 54], it is interesting to investigate whether the robustness properties of SAM can be preserved when training with SAM-ON instead of SAM-all. In Table 15 we report the adversarial robustness of the ViT-S trained from scratch on ImageNet (as reported in Section 4.2 evaluated with the two white-box attacks from APGD, but for more radii). The SAM-ON models are not only better than the base optimizer, but consistently outperform the SAM-all models by a small margin. For a WRN-28-10 on CIFAR-100 the differences are less pronounced and often within the standard deviation (reported over 3 seeds in Table 14). SAM-ON also improves over SAM-all, but for the ASAM-elementwise-$\ell_\infty$ the all-variant is slightly better than the ON-variant. Overall, we find that in order to get SAM-like improvements for adversarial robustness (as shown in [52]) it is enough to only perturb the normalization layers in SAM, illustrating again their special role.

Table 14: **Adversarial robustness CIFAR-100:** Reported is robust accuracy (in %) for a WRN-28 trained from scratch on CIFAR-100. Adversarial robustness is evaluated with the two whitebox APGD attacks from autoattack [16].

| | | SGD | SAM | | ASAM-el.-$l_\infty$ | |
|---|---|---|---|---|---|---|
| threat model | $\epsilon$ | | all | ON | all | ON |
| $\ell_2$ | 0.10 | $18.14^{\pm0.11}$ | $28.14^{\pm1.09}$ | **31.28** $^{\pm0.50}$ | $30.33^{\pm0.80}$ | $30.16^{\pm0.26}$ |
| $\ell_2$ | 0.20 | $2.33^{\pm0.11}$ | $5.39^{\pm0.34}$ | **6.62** $^{\pm0.07}$ | **6.63** $^{\pm0.12}$ | $6.10^{\pm0.18}$ |
| $\ell_\infty$ | $1/255$ | $10.29^{\pm0.04}$ | $17.96^{\pm1.08}$ | **19.56** $^{\pm0.33}$ | **20.69** $^{\pm0.81}$ | $18.63^{\pm0.30}$ |
| $\ell_\infty$ | $2/255$ | $0.67^{\pm0.01}$ | $1.96^{\pm0.17}$ | **2.16** $^{\pm0.07}$ | **2.62** $^{\pm0.01}$ | $2.05^{\pm0.17}$ |
| Clean acc. | | $80.7^{\pm0.2}$ | $83.1^{\pm0.3}$ | $\mathbf{84.2}^{\pm0.2}$ | $83.3^{\pm0.2}$ | $\mathbf{84.1}^{\pm0.2}$ |

Table 15: **Adversarial robustness ImageNet:** Reported is robust accuracy (in %) for a ViT-S trained from scratch on ImageNet, as reported in Table 4. Adversarial robustness is evaluated with the two whitebox APGD attacks from autoattack [16].

| | | AdamW | | | Lion | | |
|---|---|---|---|---|---|---|---|
| | $\epsilon$ | vanilla | SAM-all | SAM-ON | vanilla | SAM-all | SAM-ON |
| $\ell_2$ | 0.25 | $19.67^{\pm0.47}$ | $37.53^{\pm0.69}$ | **41.16** $^{\pm0.24}$ | $22.01^{\pm0.78}$ | $38.52^{\pm0.66}$ | **43.12** $^{\pm0.97}$ |
| $\ell_2$ | 0.50 | $5.47^{\pm0.18}$ | $17.71^{\pm0.61}$ | **22.72** $^{\pm0.25}$ | $6.63^{\pm0.46}$ | $19.03^{\pm0.92}$ | **24.27** $^{\pm1.34}$ |
| $\ell_2$ | 1.00 | $0.43^{\pm0.09}$ | $3.34^{\pm0.36}$ | **5.58** $^{\pm0.19}$ | $0.57^{\pm0.07}$ | $3.98^{\pm0.28}$ | **6.64** $^{\pm0.69}$ |
| $\ell_\infty$ | $0.25/255$ | $33.45^{\pm0.80}$ | $48.08^{\pm0.14}$ | **49.34** $^{\pm0.08}$ | $35.31^{\pm0.08}$ | $49.57^{\pm0.60}$ | **51.37** $^{\pm0.99}$ |
| $\ell_\infty$ | $0.5/255$ | $14.98^{\pm0.18}$ | $29.68^{\pm0.09}$ | **32.46** $^{\pm0.15}$ | $15.86^{\pm0.13}$ | $31.68^{\pm0.62}$ | **34.23** $^{\pm1.73}$ |
| $\ell_\infty$ | $1/255$ | $2.61^{\pm0.16}$ | $8.64^{\pm0.01}$ | **10.82** $^{\pm0.56}$ | $2.93^{\pm0.29}$ | $10.02^{\pm0.56}$ | **12.03** $^{\pm1.30}$ |
| Clean acc. | | $66.89^{\pm0.04}$ | $71.47^{\pm0.12}$ | $71.37^{\pm0.026}$ | $68.20^{\pm0.02}$ | $71.90^{\pm0.19}$ | $\mathbf{72.64}^{\pm0.14}$ |

### B.5 Machine translation task

To probe the effectiveness of *SAM-ON* outside the vision domain, we apply it to the IWSLT'14 DE-EN machine translation task, following the setup of Kwon et al. [37]. We report the resulting Bleu scores in Table 16: *SAM-all* and *SAM-ON* perform similar (within standard deviations reported over 3 random seeds), both improving over the vanilla optimizer. While being very limited in its scope, this experiment is a first hint that *SAM-ON* might also be effective outside the vision domain. Proper evaluations, as for instance done in [7], are required to confirm this for large-scale settings.

Table 16: IWSLT-DE-EN Bleu scores. Reported over 3 random seeds.

| vanilla | SAM-all | SAM-ON |
|---|---|---|
| $34.56^{\pm0.11}$ | $34.83^{\pm0.10}$ | $34.95^{\pm0.16}$ |

### B.6 Weight distribution after training

In order to get a better understanding of the impact of *SAM-ON* on $\gamma$ and $\beta$ (as defined in Eq. 1), we train a WideResNet-28-10 with different SAM-variants and both *SAM-ON* and *all*. We show the distribution of $|w_i|$, i.e. the parameter magnitudes, at the end of training for different layer types in Figure 9. Different to the discussion in Section 5.3, we show the $y$-axis on log-scale, in order to inspect more nuanced differences. For elementwise $\ell_2$ there is no strong change in the distribution of the BatchNorm parameters between *all* and *SAM-ON*. For elementwise $\ell_\infty$, layerwise $\ell_2$ and SAM, however, the magnitude of the BatchNorm parameters shifts clearly towards larger values, especially for the weight parameters. We note that this resembles a pattern we observed when comparing the optimal $\rho$-value for *all* and *SAM-ON* in Table 7: The optimal $\rho$ of elementwise $\ell_2$ did not change much for ResNet architectures, whereas for the other considered methods, it shifted towards larger values for *SAM-ON*. Additionally and in contrast to the other methods, the elementwise $\ell_2$ variant showed a strong performance decrease in *no-norm* (Figure 1), indicating that it implicitly focuses on perturbing the BatchNorm layers already. We note that larger BatchNorm parameters do not necessarily indicate a functionally different network, since there are many reparameterization invariances in ReLU networks, some of which ASAM tries to leverage in its perturbation definition Eq. (4). Nevertheless, the scale of the network still has an impact on the training dynamics, since other methods like e.g. weight decay depend on it. We discuss the impact of weight decay further in Appendix B.9.

### B.7 Removing the affine parameters

Frankle et al. [24] found for SGD that fixing the normalization parameters typically decreases the generalization performance of networks. As an ablation, we therefore study the effect of SAM when the normalization weights are non-trainable. This is, we set $\gamma = 1$ and $\beta = 0$ and train the remaining parameters with SAM. The results are shown for a WRN-28 in Figure 10, where it can be seen that fixing the normalization parameters (*fix-norm*) does *not* lead to a decrease in the performance of SAM. We thus hypothesize that in certain settings, SAM might not leverage the expressive power of the normalization layers, which might contribute to the improved performance of *SAM-ON*.

### B.8 Training BatchNorm and only BatchNorm

The affine parameters of the normalization layers are relatively understudied in the literature. Recently, Frankle et al. [24] were able to obtain surprisingly high performance for ResNet architectures by only training the BatchNorm layers (freezing all other parameters), illustrating their expressive power. We study the effect of SAM in this setting (i.e. when all parameters except for the BatchNorm layers are frozen) for a ResNet-101 and a WRN-28 on CIFAR-10 and find that SAM still aids generalization in this setting (Table 17).

Table 17: Effect of SAM when training *only* BatchNorm layers, for networks trained on CIFAR-10.

| Model | SGD | SAM $\rho = 0.01$ | SAM $\rho = 0.05$ |
|---|---|---|---|
| ResNet-101 | 78.75 | 78.63 | **79.27** |
| WRN-28 | 63.49 | **64.48** | 62.70 |

### B.9 Weight decay and dropout

Here, we explore potential connections of *SAM-ON* with weight decay and dropout. Since weight decay is sometimes applied to all network parameters, and sometimes normalization layers are omitted, it is worth investigating if the benefits of *SAM-ON* can be attributed to its interaction with weight decay. To this end, we train a WRN-28 with SGD, *SAM-all* and *SAM-ON*, and apply weight decay to either all parameters, all *except* the normalization layers, or not at all (Figure 11, right). For each setting *SAM-ON* outperforms SAM, outlining that its success should not be attributed to the interaction with weight decay.

We further test if *SAM-ON*-like performance can be achieved by simply applying stronger regularization and stochasticity to the normalization parameters. To this end, we apply dropout solely on the normalization layers (Figure 11, left) and find that this is not the case.

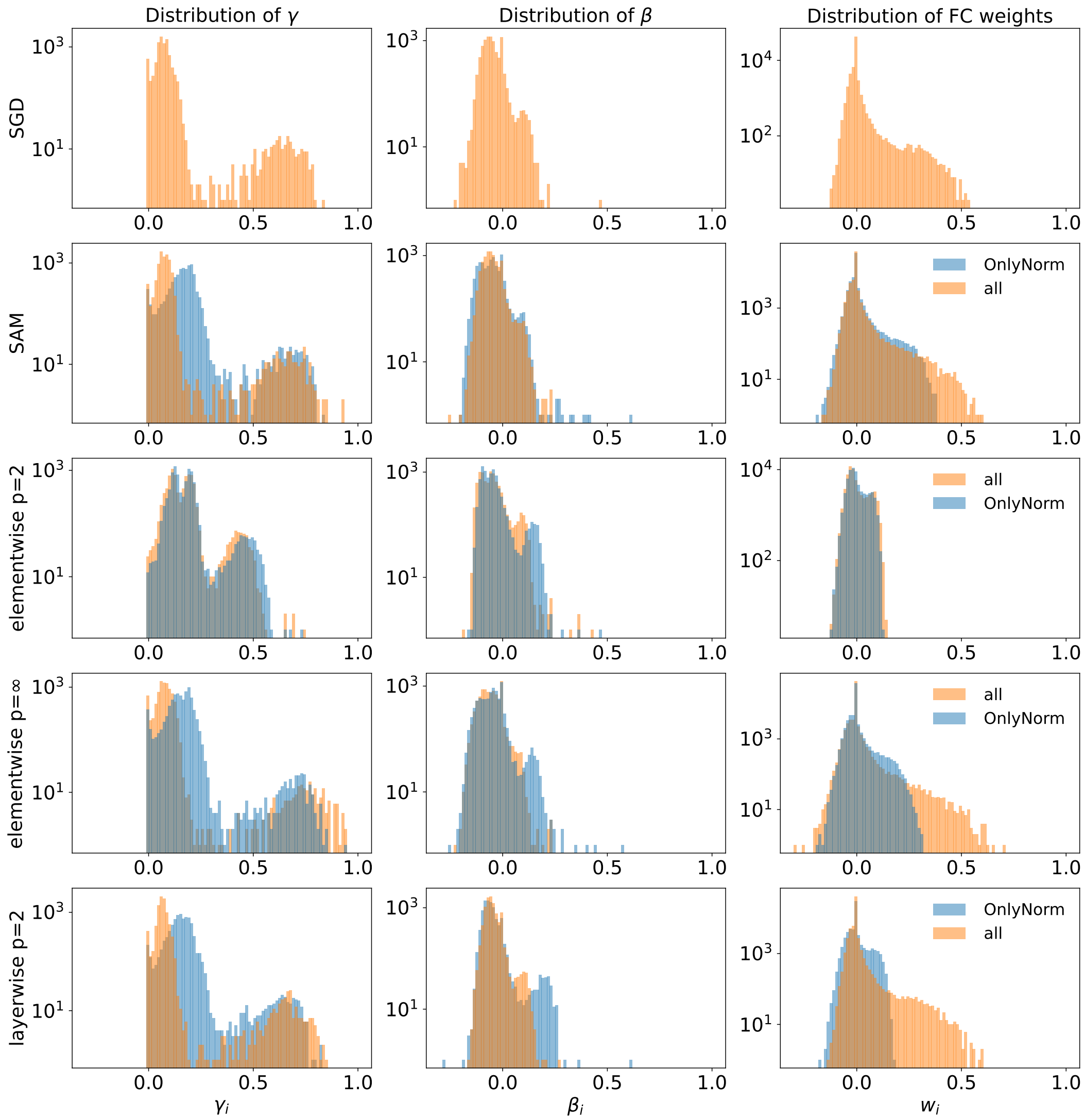

Figure 9: SAM-ON leads to a shift in the distribution of $\gamma$.

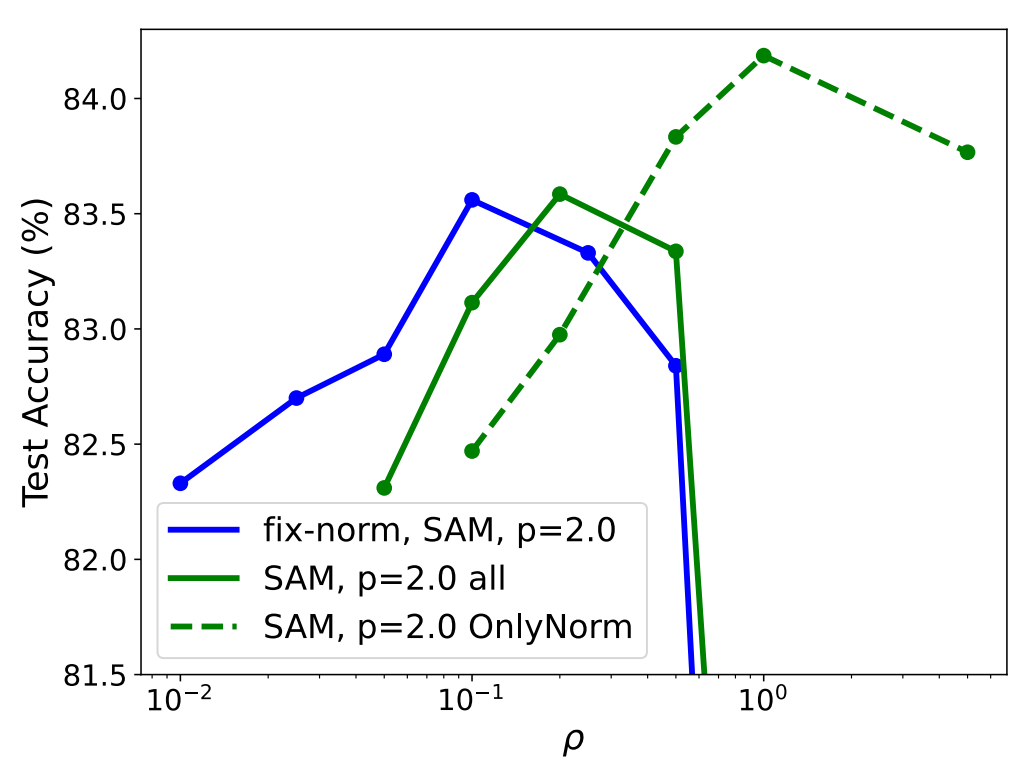

Figure 10: When training with SAM, fixing $\gamma = 1, \beta = 0$ (*fix-norm*) barely changes the performance of the network. WRN-28, CIFAR-100.

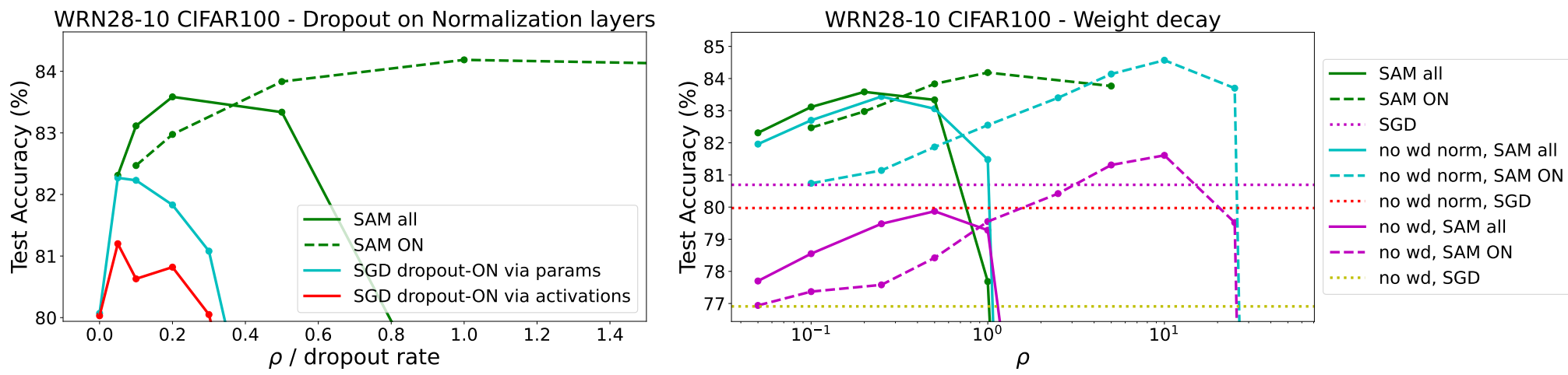

Figure 11: **Left:** Applying dropout only to the normalization layers (blue/red) performs worse than *SAM-ON*. **Right:** *SAM-ON* improves over *SAM-all* irrespective of whether weight decay is applied to all parameters (green), all *except* the normalization layers (blue) or not at all (yellow).

### B.10 Details on sharpness evaluation

For the following discussion, we note that the term *generalization* is sometimes used as the difference between train and test error, while in other cases people use it as a synonym for test error. Since in CIFAR settings the models achieve train error close to zero, the two definitions become equivalent.

Many studies have attempted to better understand the possible connection between the generalization of deep neural networks and the flatness of the loss-surface [26, 32, 22, 34, 18]. Recently, Andriushchenko et al. [4] conducted a large-scale study for a range of models, datasets, and sharpness-definitions, finding that "while there definitely exist restricted settings where correlation between sharpness and generalization is significantly positive (e.g., for ResNets on CIFAR-10 with a specific combination of augmentations and mixup) it is not true anymore when we compare all models jointly" and concluding "that one should avoid blanket statements like *flat minima generalize better*". In order to evaluate sharpness, we therefore adopt their setup and choose the best-performing sharpness measure for CIFAR from their study, which is logit-normalized elementwise-adaptive worst-case-$\ell_\infty$-$m$-sharpness. This is, $m$-sharpness $s_w^m$ is defined as the largest possible change in loss within the adaptive perturbation model defined in 4,

$$s_w^m = \mathbb{E}_{x,y\sim D_m} \max_{||T_w^{-1}\epsilon||_p \leq \rho} L(\mathbf{w}+\epsilon) - L(\mathbf{w}) \tag{6}$$

where $T_w^i = |w_i|$, $p = \infty$ and $D_m$ returns data batches of size $m$. $\rho$ here denotes the size of the ball over which sharpness is evaluated and is not to be confused with the $\rho$ from the SAM-algorithm. Like for ASAM [37], the motivation behind adaptive sharpness measures is to make them invariant to reparameterizations of the network. Further, the logit-outputs of the network are normalized with respect to their $\ell_2$-norm in order to mitigate the scale-sensitivity of classification losses. In practice, Andriushchenko et al. [4] compute $s_w^m$ over a subset of the train set of size $1024$ and use $m = 128$, i.e. average 8 batches. We use a subset of size $2048$ in order to obtain more reliable sharpness estimates, and adopt $m = 128$. The maximization in (6) is performed with AutoPGD [16], a hyperparameter-free method designed for accurate estimation of adversarial robustness. It is to note that except for the logit-normalization, the sharpness definition reported in Table 6 corresponds exactly to the perturbation model that ASAM elementwise $\ell_\infty$ uses, and hence the 1-step sharpness reported should be fairly close the the objective that ASAM elementwise $\ell_\infty$ actually minimizes during training. While ASAM elementwise $\ell_\infty$ yields slightly smaller sharpness values than the conventional SAM algorithm, the differences are rather small when compared to the significantly sharper *SAM-ON* models. For the results in Table 6 in the main paper we tuned the sharpness radius $\rho$ such that we obtain sharpness values similar to those reported to yield the highest correlation in Andriushchenko et al. [4]. In Table 18 we report sharpness values for a ResNeXt-model, in addition to the WRN-28 from the main paper. In all cases the *SAM-ON* models are sharper than the *SAM-all* models yet generalize better. In Table 19 we further report other sharpness measures without logit-normalization for a WRN-28. SAM-ON is sharper than SAM-all with respect to most metrics, although there exist some exceptions. It should however be stressed that many of those metrics did not show good correlation with generalization in the study by Andriushchenko et al. [4].

Table 18: Sharpness evaluation of both a WRN-28 and a ResNeXt. *SAM-ON* is sharper than *SAM-all* in all cases. Shown is 20-step logit-normalized $\ell_\infty$ sharpness from [2], averaged over three models per method. Dataset considered is CIFAR-100.

| | | SGD | SAM | | ASAM-el.-$\ell_\infty$ | |
|---|---|---|---|---|---|---|
| | | | all | ON | all | ON |
| WRN-28 | Test Accuracy (%) | $80.71^{\pm 0.2}$ | $83.11^{\pm 0.3}$ | $\mathbf{84.19}^{\pm 0.2}$ | $83.25^{\pm 0.2}$ | $\mathbf{84.14}^{\pm 0.2}$ |
| | $\ell_\infty$-sharpness, $\rho = 0.003$ | $0.071^{\pm 0.000}$ | $\mathbf{0.048}^{\pm 0.001}$ | $0.090^{\pm 0.005}$ | $\mathbf{0.048}^{\pm 0.001}$ | $0.078^{\pm 0.004}$ |
| | $\ell_\infty$-sharpness, $\rho = 0.005$ | $0.201^{\pm 0.001}$ | $\mathbf{0.139}^{\pm 0.004}$ | $0.296^{\pm 0.018}$ | $\mathbf{0.124}^{\pm 0.002}$ | $0.283^{\pm 0.011}$ |
| | $\ell_\infty$-sharpness, $\rho = 0.007$ | $0.433^{\pm 0.002}$ | $\mathbf{0.309}^{\pm 0.011}$ | $0.585^{\pm 0.018}$ | $\mathbf{0.255}^{\pm 0.005}$ | $0.580^{\pm 0.020}$ |
| ResNeXt | Test Accuracy (%) | $80.16^{\pm 0.3}$ | $81.79^{\pm 0.4}$ | $\mathbf{82.22}^{\pm 0.2}$ | $81.02^{\pm 0.6}$ | $\mathbf{82.38}^{\pm 0.3}$ |
| | $\ell_\infty$-sharpness, $\rho = 0.001$ | $0.036^{\pm 0.001}$ | $\mathbf{0.029}^{\pm 0.000}$ | $0.034^{\pm 0.000}$ | $\mathbf{0.026}^{\pm 0.002}$ | $0.034^{\pm 0.001}$ |
| | $\ell_\infty$-sharpness, $\rho = 0.003$ | $0.164^{\pm 0.005}$ | $\mathbf{0.117}^{\pm 0.004}$ | $0.140^{\pm 0.002}$ | $\mathbf{0.099}^{\pm 0.010}$ | $0.147^{\pm 0.001}$ |
| | $\ell_\infty$-sharpness, $\rho = 0.005$ | $0.383^{\pm 0.011}$ | $\mathbf{0.252}^{\pm 0.008}$ | $0.291^{\pm 0.005}$ | $\mathbf{0.203}^{\pm 0.021}$ | $0.312^{\pm 0.001}$ |

Table 19: Additional sharpness measures. WRN-28 (no logitnorm).

| | | SGD | SAM | | ASAM-el.-$\ell_\infty$ | |
|---|---|---|---|---|---|---|
| | adaptive | | all | ON | all | ON |
| Test Accuracy (%) | | $80.71^{\pm 0.2}$ | $83.11^{\pm 0.3}$ | $\mathbf{84.19}^{\pm 0.2}$ | $83.25^{\pm 0.2}$ | $\mathbf{84.14}^{\pm 0.2}$ |
| $\ell_2$ avg, $\rho = 0.005$ | False | $1.358^{\pm 0.049}$ | $\mathbf{0.515}^{\pm 0.020}$ | $2.372^{\pm 0.071}$ | $\mathbf{0.569}^{\pm 0.012}$ | $2.141^{\pm 0.045}$ |
| $\ell_2$ avg, $\rho = 0.1$ | True | $0.042^{\pm 0.001}$ | $\mathbf{0.019}^{\pm 0.001}$ | $0.022^{\pm 0.001}$ | $0.040^{\pm 0.001}$ | $\mathbf{0.019}^{\pm 0.001}$ |
| $\ell_\infty$ avg, $\rho = 0.01$ | False | $2.643^{\pm 0.097}$ | $\mathbf{1.264}^{\pm 0.028}$ | $3.455^{\pm 0.050}$ | $\mathbf{1.304}^{\pm 0.007}$ | $3.259^{\pm 0.031}$ |
| $\ell_\infty$ avg, $\rho = 0.2$ | True | $0.078^{\pm 0.001}$ | $0.035^{\pm 0.001}$ | $0.034^{\pm 0.004}$ | $0.068^{\pm 0.003}$ | $\mathbf{0.031}^{\pm 0.001}$ |
| $\ell_2$-worst, $\rho = 0.05$ | False | $0.501^{\pm 0.048}$ | $0.655^{\pm 0.277}$ | $0.701^{\pm 0.057}$ | $0.768^{\pm 0.141}$ | $\mathbf{0.313}^{\pm 0.044}$ |
| $\ell_2$-worst, $\rho = 0.25$ | True | $0.065^{\pm 0.008}$ | $0.033^{\pm 0.004}$ | $0.037^{\pm 0.017}$ | $0.056^{\pm 0.006}$ | $0.062^{\pm 0.001}$ |
| $\ell_\infty$-worst, $\rho = 1e-05$ | False | $0.149^{\pm 0.003}$ | $\mathbf{0.055}^{\pm 0.002}$ | $0.144^{\pm 0.005}$ | $\mathbf{0.050}^{\pm 0.002}$ | $0.123^{\pm 0.007}$ |
| $\ell_\infty$-worst, $\rho = 0.004$ | True | $0.537^{\pm 0.023}$ | $\mathbf{0.262}^{\pm 0.009}$ | $0.600^{\pm 0.053}$ | $\mathbf{0.255}^{\pm 0.011}$ | $0.505^{\pm 0.027}$ |

# C Convergence Analysis

We provide in this section a convergence analysis for *SAM-ON* in the non-convex setting. Using standard assumptions we obtain a theorem which resembles findings for closely related methods such as found in [2, 42].

Our assumptions:

**Assumption C.1.** We assume function $f : \mathbb{R}^n \to \mathbb{R}$ to be $L$-smooth: there exists $L > 0$ such that

$$\|\nabla f(v) - \nabla f(w)\|_2 \leq L\|v - w\|_2, \ \forall v, w \in \mathbb{R}^n. \tag{7}$$

**Assumption C.2.** There exists $M > 0$ for any sample $x_i$ such that

$$\|\nabla f_{x_i}(w)\|_2^2 \leq M, \ \ \forall w \in \mathbb{R}^n. \tag{8}$$

*Remark* C.3. If Assumption C.1 holds ($L$-smoothness), then $\forall v, w \in \mathbb{R}^n$:

$$|f(v) - (f(w) + \nabla f(w)^T(v - w))| \leq \frac{L}{2}\|v - w\|_2^2. \tag{9}$$

This well-known result can be derived using the fundamental theorem of calculus and Cauchy-Schwartz.

*Remark* C.4. Assumption C.2 guarantees that the variance of the stochastic gradient is less than $M$.

*SAM-ON.* In the following we shall denote the true gradient as $\nabla f(w)$ and the noisy observation gradient as $g(w)$. The gradient of the loss of the $i$th training example is denoted as $g_{x_i}(w)$. We partition the neural network parameters layer-wise as $w = \{w_N, w_A\}$, with $w_N \in \mathbb{R}^{n_N}, w_A \in \mathbb{R}^{n_A}$, $n = n_N + n_A$, where $w_N$ represent the normalization layer parameters and $w_A$ all other layers. The

iteration for $w_N$ is:

$$w_N^{t+1/2} = w_N^t + \rho \frac{g_{N,x_i}(w^t)}{\|g_{N,x_i}(w^t)\|}$$
$$w_N^{t+1} = w_N^t - h\, g_{N,x_i}\left(w^{t+1/2}\right) \tag{10}$$

and for $w_A$ is:

$$w_A^{t+1/2} = w_A^t$$
$$w_A^{t+1} = w_A^t - h\, g_{A,x_i}\left(w^{t+1/2}\right). \tag{11}$$

**Theorem C.5.** *Assuming C.1 and C.2, $h \leq 1/L$, we obtain:*

$$\frac{1}{T}\sum_{t=0}^{T-1} \mathbb{E}\left[\|\nabla f(w^t)\|^2\right] \leq \frac{2(f(w^0) - f(w^*))}{hT} + 2LhM + L^2\rho^2(1 + Lh), \tag{12}$$

*with $w^*$ the optimal solution to $f(w)$.*

*Proof.* From Assumption C.1 and thus Remark C.3 it follows that:

$$\begin{aligned}
f(w^{t+1}) &\leq f(w^t) + \nabla f(w^t)\cdot(w^{t+1} - w^t) + \frac{L}{2}\|w^{t+1} - w^t\|^2 && (13)\\
&\leq f(w^t) - h\nabla f(w^t)\cdot g_{x_i}\left(w^{t+1/2}\right) + \frac{h^2 L}{2}\left\|g_{x_i}\left(w^{t+1/2}\right)\right\|^2 && (14)\\
&= f(w^t) - h\nabla f(w^t)\cdot g_{x_i}\left(w^{t+1/2}\right)\\
&\quad + \frac{h^2 L}{2}\left(\|\nabla f(w^t) - g_{x_i}(w^{t+1/2})\|^2 - \|\nabla f(w^t)\|^2 + 2\left(\nabla f(w^t)\cdot g_{x_i}(w^{t+1/2})\right)\right)\\
&= f(w^t) - \frac{Lh^2}{2}\|\nabla f(w^t)\|^2 + \frac{Lh^2}{2}\|\nabla f(w^t) - g_{x_i}(w^{t+1/2})\|^2\\
&\quad - (1 - Lh)h\left(\nabla f(w^t)\cdot g_{x_i}(w^{t+1/2})\right) && (15)\\
&\leq f(w^t) - \frac{Lh^2}{2}\|\nabla f(w^t)\|^2 + Lh^2\|\nabla f(w^t) - g_{x_i}(w^t)\|^2\\
&\quad + Lh^2\|g_{x_i}(w^t) - g_{x_i}(w^{t+1/2})\|^2 - (1 - Lh)h\left(\nabla f(w^t)\cdot g_{x_i}(w^{t+1/2})\right). && (16)
\end{aligned}$$

Taking the double expectation gives (because unbiased gradient and Assumption C.2 and Remark C.4):

$$\begin{aligned}
\mathbb{E}[f(w^{t+1})] \leq \mathbb{E}[f(w^t)] - \frac{Lh^2}{2}\mathbb{E}\|\nabla f(w^t)\|^2 + Lh^2 M\\
+ \underbrace{Lh^2\|g(w^t) - g(w^{t+1/2})\|^2}_{\mathcal{A}} - (1 - Lh)h\underbrace{\mathbb{E}\left[\nabla f(w^t)\cdot g(w^{t+1/2})\right]}_{\mathcal{B}}. && (17)
\end{aligned}$$

For term $\mathcal{A}$ we obtain using Assumption C.1:

$$\mathcal{A} \leq L^3 h^2 \|w^t - w^{t+1/2}\|^2 = L^3 h^2 \rho^2. \tag{18}$$

For term $\mathcal{B}$ we obtain:

$$\begin{aligned}
\mathcal{B} &= \mathbb{E}\left[\{\nabla f_N(w^t), \nabla f_A(w^t)\}\cdot\{g_N(w^{t+1/2}), g_A(w^{t+1/2})\}\right] && (19)\\
&= \mathbb{E}[\nabla f_A(w^t)\cdot(g_A(w^{t+1/2}) - g_A(w^t) + g_A(w^t))]\\
&\quad + \mathbb{E}[\nabla f_N(w^t)\cdot(g_N(w^{t+1/2}) - g_N(w^t) + g_N(w^t))] && (20)\\
&= \mathbb{E}\left[\|\nabla f(w^t)\|^2\right]\\
&\quad + \underbrace{\mathbb{E}[\nabla f_A(w^t)\cdot(g_A(w^{t+1/2}) - g_A(w^t))] + \mathbb{E}[\nabla f_N(w^t)\cdot(g_N(w^{t+1/2}) - g_N(w^t))]}_{\mathcal{C}}. && (21)
\end{aligned}$$

Using $xy \leq \frac{1}{2}\|x\|_2^2 + \frac{1}{2}\|y\|_2^2$ and Assumption C.1 we get for $\mathcal{C}$:

$$|\mathcal{C}| \leq \frac{1}{2}\mathbb{E}\left[\|\nabla f(w^t)\|^2\right] + \frac{L^2}{2}\|w^{t+1/2} - w^t\|^2 = \frac{1}{2}\mathbb{E}\left[\|\nabla f(w^t)\|^2\right] + \frac{L^2\rho^2}{2}. \tag{22}$$

Plugging this into (17) gives:

$$\begin{aligned}
\mathbb{E}[f(w^{t+1})] \leq \mathbb{E}[f(w^t)] - \frac{Lh^2}{2}\mathbb{E}\|\nabla f(w^t)\|^2 + Lh^2 M + L^3h^2\rho^2 - (1-Lh)h\mathbb{E}\|\nabla f(w^t)\|^2 \\
+ (1-Lh)h\left(\frac{1}{2}\mathbb{E}\|\nabla f(w^t)\|^2 + \frac{L^2\rho^2}{2}\right) \qquad (23) \\
\leq \mathbb{E}[f(w^t)] - \frac{h}{2}\mathbb{E}\|\nabla f(w^t)\|^2 + Lh^2 M + \frac{1}{2}hL^2\rho^2(1+Lh). \qquad (24)
\end{aligned}$$

In $T$ iterations we obtain using a telescoping sum:

$$\begin{aligned}
f(w^*) - f(w^0) &\leq \mathbb{E}[f(w^T)] - f(w^0) \\
&\leq -\frac{h}{2}\sum_{t=0}^{T-1}\mathbb{E}\left[\|\nabla f(w^t)\|^2\right] + Lh^2MT + \frac{1}{2}hL^2\rho^2(1+Lh)T. \qquad (25)
\end{aligned}$$

This gives Theorem C.5. □