# OpenReview forum: "Normalization Layers Are All That Sharpness-Aware Minimization Needs"
_NeurIPS.cc/2023/Conference — NeurIPS 2023 poster_

### Official Review · Reviewer_sERj · 2023-06-27

**Soundness:** 3 good
**Presentation:** 2 fair
**Contribution:** 2 fair
**Rating:** 3
**Confidence:** 4

**Summary:**

The paper proposes an adversarial perturbation method linked to SAM for the affine normalization parameters, in contrast to perturbing the full set of parameters. The results show this approach improves upon standard SAM and prior sparse SAM approaches.

**Strengths:**

The SAM-ON proposed method yields several benefits:
1. improves upon SAM-all and carious other SAM-variants
2. achieves this by perturbing only the normalization layers, which correspond to a small percentage of the total parameters


**Weaknesses:**

Weaknesses of this paper include:
1. gains in Table 1 are marginal
2. originality of approach is lacking given prior literature on SAM variants
3. justification of SAM-ON is poor; the experiments in 5.2 and Table 7 try to poke at this but fall short as the authors do not provide a solid explanation as to what this can be attributed to; they only show that SAM-ON increases sharpness in L-inf sense; but what about other metrics of flatness? is there a more universal metric that can explain the effectiveness of SAM-ON?
4. it is unclear whether SAM-ON is truly better than other SAM variants in out of distribution robustness, e.g. WILDS benchmark
5. results are only shown for CIFAR and ImageNet datasets; more datasets and benchmarks need to be considered

**Questions:**

see weaknesses above

One of the questions this paper fails to answer is why sharpness is higher for SAM-ON, even though its formulation is motivated to minimize sharpness. This is not a complete enough story and opens up lots of questions that need to be addressed.

**Limitations:**

Somewhat discussed but limited. Would like to see a more thorough limitations discussion.

---

> ### Author Rebuttal · Authors · 2023-08-09
>
> Thank you for your review. Following your suggestion, we are happy to extend the limitations section of the paper. In case there is specific work you think would need to be included, we would much appreciate a pointer. Below we address your other comments.
>
> 1. __“gains in Table 1 are marginal”__:
> SAM-ON consistently strongly outperforms base optimizers and its gains compared to SAM are often >1% for ResNet architectures on CIFAR data. Similar or no gains were also observed in other SAM variant papers, such as ASAM and FisherSAM, so the amount of gain observed is within expectations. Further, we provide additional experiments on VGG-Nets and DenseNet for CIFAR data in Table 1 in the rebuttal pdf and observe that SAM-ON clearly outperforms SAM-all. Finally, our gains are even more impressive for ViTs (Table 3, main paper), which we believe to be very important as SAM has shown particular success in this domain (Chen et al., 2022).
>
> 2. __“originality of approach is lacking”__:
> To illustrate the originality of our approach (as also corroborated by reviewer dJUQ), we briefly recap here the most closely related works and how they differ from our approach. The most related works are by: 1) Frankle et al., 2021, which specifically studies the BatchNorm parameters by only training BatchNorm, but in a very different setting: without SAM and all other parameters are frozen. 2) Mi et al., 2022 which propose a sparse SAM approach, but again the approach and aim is very different: they do not highlight the importance of specific parameter groups, whereas we stress the unique role played by the normalization layers allowing for an extremely sparse yet effective perturbation, and their aim is to reduce the computational cost, not to enhance generalization (we outperform them in both). 3) Much of the literature on SAM proposes new variants based on designing better perturbation models by taking reparametrization invariances (like ASAM) or loss geometry perspectives (like Fisher-SAM) into account. We think that the fact that our simple heuristic “perturb only the normalization layers” aids all of those methods is quite remarkable. We hope this addresses the reviewer’s concerns. If the reviewer is aware of further literature relating SAM to the normalization layers of a network we would much appreciate a specific pointer.
>
> 3. __“other metrics of flatness”__:
> Andriushchenko et al., 2023 performed a large-scale study on the relation between a range of sharpness measures and generalization, finding little correlation. For CIFAR data, "_the best correlation [...] is achieved by_ $\ell_\infty$ _adaptive worst-case sharpness with logit normalization for a small_ $\rho$". Since this metric is additionally very close to the perturbation model of ASAM elementwise $\ell_\infty$ we chose to focus on this particular metric in our paper. Following your suggestion, we investigated more sharpness measures (Table 3 in rebuttal pdf) and find in alignment with our previous findings that SAM-ON is sharper than SAM-all with respect to most metrics (especially worst-case sharpness and the metrics that are optimized during SAM), although there exist some exceptions. Results for more models and $\rho$ values are similar (omitted due to space limitations) and will be included in the revised paper.
>
> 4. __“a universal metric that can explain the effectiveness of SAM-ON”__
> There is a long history of trying to find metrics that correlate well with generalization performance. One of these is sharpness, which we focused on in this paper. Our work lends evidence to recent findings (Andriushchenko et al., 2023) that sharpness may not always correlate well with generalization performance. The search for alternative explanations for SAM’s success is an active area of research, for example recent work (that appeared on arXiv after the NeurIPS submission deadline and hence not included in our related work section) suggests that the use of SAM prunes a significant number of activations and leads to low-rank features (arXiv:2305.16292). Our work contributes to this search for understanding the fundamental reasons for the success of SAM by illustrating the important role played by the normalization layers.
>
> 5. __“OOD robustness”__:
> In Table 2 in the rebuttal pdf we have provided experiments for training Vision Transformers from scratch on ImageNet using SAM-all/SAM-ON with base optimizers AdamW and Lion, and evaluated those on OOD test sets (ImageNet-Sketch, ImageNet-R). We find consistent improvements of SAM-ON over SAM-all on ImageNet, ImageNet-Sketch, and ImageNet-R.
>
> 6. __“more datasets and benchmarks”__:
> For the selection of our datasets and benchmarks we followed the literature on SAM and its variants. Additionally, we include in the rebuttal pdf (Table 1 and 2) results for OOD robustness as suggested by the reviewer, ViTs trained from scratch on ImageNet as suggested by reviewer Ftd6, and more models on CIFAR as suggested by reviewer wExQ. For all of these we observe consistent gains of SAM-ON over SAM-all. We hope this addresses your concern, but do let us know if there is anything that is not adequately supported.
>
> 7. __“why is sharpness is higher for SAM-ON, even though its formulation is motivated to minimize sharpness.”__
> Opinions differ on whether our findings that SAM-ON may increase sharpness are trivially true or unexpected, e.g. as mentioned by Reviewer wExQ: “The observation that .. [SAM-ON] .. holds the higher sharpness than original SAM might be trivial. SAM are designed to reduce the sharpness, while a degraded version of SAM should hold a higher sharpness in principle.” This is exactly why we believe it is important to study sharpness in our work because it is not obvious how we would expect it to behave with respect to SAM-ON. Our findings lend support to the idea that sharpness may not be the sole reason for SAM’s success and aid the search for alternative explanations (Andriushchenko et al., 2023).

---

> > ### Comment · Reviewer_sERj · 2023-08-13
> >
> > I thank the authors for their rebuttals. While I understand the recent paper "Towards Understanding Sharpness-Aware Minimization" also notes that convergence to flat minima with SAM is incomplete, this paper does not move the needle forward much. There is a lack of theory and metrics related to the normalization layers that support the proposed method SAM-ON. In my view, there is still a lack of a solid reasoning about the contributions of this paper. I choose to keep my score for now.

---

> > > ### Author Response · Authors · 2023-08-15
> > >
> > > Thank you for your response.
> > >
> > >
> > > We are disappointed that the new experiments that you asked for (OOD robustness, other sharpness metrics, more benchmarks) and our explanations (table 1 gains, originality, universal metric, sharpness) are not acknowledged or discussed in your new response.
> > > With regards to your new comment: we emphasize that our contribution is fundamentally different from the study of _"Towards Understanding Sharpness-Aware Minimization"_. The latter shows that the better generalization of SAM or ASAM is mostly not correlated with finding models with flat minima. Our work highlights the important role played by the normalization layers: we show that perturbing only the normalization layers (less than 0.1% of the total parameters) enhances or at least maintains performance for SAM and all other SAM variants we considered. This is complementary to the paper _"Towards Understanding Sharpness-Aware Minimization"_  by working towards an alternative explanation on why SAM and its variants work.
> > > We would greatly appreciate it if you could elaborate on what you consider a _"lack of solid reasoning about the contributions of this paper"_ so we can engage in a discussion.

---

### Official Review · Reviewer_dJUQ · 2023-07-06

**Soundness:** 4 excellent
**Presentation:** 4 excellent
**Contribution:** 4 excellent
**Rating:** 9
**Confidence:** 4

**Summary:**

As Sharpness-Aware Minimization (SAM) aims to regularize the flatness of the loss landscape for better generalization, this paper shows that only perturbing the normalization layers is sufficient to achieve this. To prove this, the authors first propose a method called SAM-ON (SAM-OnlyNorm). Then, they conduct experiments on CIFAR datasets across different models and variants of SAM, comparing the test accuracy with and without applying SAM-ON. The experiments (in most cases) show that SAM-ON outperforms vanilla SAM, as well as SAM variants with ON outperforming SAM variants. Furthermore, since SAM-ON only needs to perturb a few parameters, it can save computational resources. Finally, this paper conducts numerous experiments to understand SAM-ON and concludes that sharpness may not be the key to SAM.

## post-rebuttal
I've updated my score since my concerns are adequately addressed.

**Strengths:**

1. This paper demonstrates a new understanding of the underlying mechanism of SAM, which is both novel and informative.
2. The proposed SAM-ON can improve efficiency and generalization in most cases.
3. The experiments are adequate to support the claims.
4. The code is provided, which guarantees reproducibility.

**Weaknesses:**

1. SAM-ON cannot outperform SAM (or its variants) in some cases, showing that SAM-ON may not always be effective and decreasing the soundness of the claims that support SAM-ON.
2. The improvement of SAM-ON on ImageNet seems to be consistently lower than on CIFAR datasets for different architectures. Therefore, this reviewer hypothesizes that SAM-ON only works for small datasets and is not very effective for large datasets.
3. As the author acknowledges, this paper only investigates SAM and SAM on vision data. Thus, the analysis cannot support the proposed understanding that generalizes to other tasks, such as language tasks.

**Questions:**

See Weaknesses.

---

> ### Author Rebuttal · Authors · 2023-08-09
>
> Thank you for your review and helpful suggestions.
> We address your concerns below.
>
> 1. __SAM-ON may not always be effective__
>    We considered a wide range of different models (see Table 1 in rebuttal pdf for even more models), data augmentations, and datasets. For almost all of these a SAM-ON variant obtained the best test accuracy. There are indeed some exceptions, but even in those settings we find it remarkable that by only applying SAM to the normalization layers, this approach can so strongly outperform base optimizers. We believe this is a valuable contribution to the literature in the search for an enhanced understanding on the mechanisms behind SAM’s success.
>
> 3. __The improvement of SAM-ON on ImageNet seems to be lower than on CIFAR datasets__
>    Additional results for ViTs trained from scratch on ImageNet can be found in Table 2 in the rebuttal pdf. We observe that SAM-ON outperforms SAM in this setting for both the AdamW and Lion optimizers, and both with respect to the ImageNet test set and OOD test sets. Further, we agree that the results for ImageNet and ResNet are less impressive than for ViT, but would like to highlight that other SAM variants such as SSAM, ESAM, GSAM, and FisherSAM also obtain only marginal (or no) improvements in this setting, so this is somewhat within expectations. We hope this addresses your concern.
>
> 5. __language tasks__
>     Thank you for the suggestion. We focused on vision data following the trend in the literature on SAM and its variants. However, a few papers do indeed study language tasks and we would be happy to include an experiment on e.g. the IWSLT’14 DE-EN machine translation task that was considered in the ASAM paper, in the camera-ready version. Please do not hesitate to let us know if there were specific experiments you had in mind.

---

> > ### Comment · Reviewer_dJUQ · 2023-08-10
> >
> > Dear Authors,
> >
> > Thanks for the rebuttal. I believe that most of my concerns have been addressed, and I would like to raise my score when review editing is allowed.
> >
> > Here are some further questions for curious.
> >
> > In the context of adversarial robustness, a recent work aims to understand SAM as implicit adversarial training and revealed that SAM is beneficial for improving robustness [1]. On the other hand, there are several works attributing adversarial vulnerabilities of DNNs through normalization layers, and designed normalization layer-based adversarial training methods to improve adversarial robustness [2,3].
> >
> > My questions are:
> >
> > 1. Can SAM-ON and the corresponding observations in this paper be the connection between normalization layers and adversarial robustness?
> > 2. Can using SAM-ON only improve adversarial robustness?
> >
> > Feel free to dismiss these questions if you are not familiar with adversarial robustness. However, I believe discussing these questions in this paper can help you strengthen the understanding of the effectiveness of SAM-ON.
> >
> > [1] Sharpness-Aware Minimization Alone can Improve Adversarial Robustness. ICML 2023 workshop
> >
> > [2] Intriguing properties of adversarial training at scale. ICLR 2020
> >
> > [3] Batch Normalization Increases Adversarial Vulnerability and Decreases Adversarial Transferability: A Non-Robust Feature Perspective. ICCV 2021

---

> > > ### Author Response · Authors · 2023-08-11
> > >
> > > Dear Reviewer,
> > >
> > > Thank you very much. We are grateful for your appreciation of our paper and for your valuable suggestions and references related to adversarial robustness. This is definitely something that we are keen to explore. Following your suggestion, we will start experiments to investigate SAM-ON with respect to adversarial robustness. We will try to include the results and a discussion on potential insights into the connection between SAM, the normalization layers, and adversarial robustness, into the revised paper. Thank you again for bringing these references to our attention.

---

> > > > ### Comment · Reviewer_dJUQ · 2023-08-11
> > > >
> > > > Dear author,
> > > >
> > > > Thanks for the reply. I have updated my score.
> > > >
> > > > Looking forward to the discussion and experimental results of my questions.
> > > >
> > > > Best,

---

> > > > > ### Author Response · Authors · 2023-08-18
> > > > >
> > > > > Dear Reviewer,
> > > > >
> > > > > Following your suggestion, we have performed a preliminary study on the adversarial robustness of SAM-all and SAM-ON (with a WRN28-10 on CIFAR100). We find that both SAM-all and SAM-ON can significantly improve over SGD-trained models with respect to adversarial robustness. Interestingly, SAM-ON is slightly more robust than SAM-all, whereas ASAM-elem.-$\ell_\infty$-all is slightly more robust than ASAM-elem.-$\ell_\infty$-ON, but the differences are small and often within the standard deviation (reported over 3 seeds). Overall, we find that in order to get SAM-like improvements for adversarial robustness (as shown in https://arxiv.org/pdf/2305.05392.pdf) it is enough to only perturb the normalization layers in SAM, illustrating again their special role as outlined in our paper. These results are preliminary, but we found them interesting enough to report them already. Thank you again for the suggestion!
> > > > >
> > > > >
> > > > > | threat model                     | $\epsilon$ | SGD                 | SAM-all             | SAM-ON                       | ASAM-el.-$l_\infty$-all      | ASAM-el.-$l_\infty$-ON      |
> > > > > |----------------------------------|------------|---------------------|---------------------|------------------------------|------------------------------|-----------------------------|
> > > > > | $\ell_2$                         | $0.10$     | $18.14 ^{\pm 0.11}$ | $28.14 ^{\pm 1.09}$ | $	{31.28 }^{\pm 0.50}$  | ${30.33 }^{\pm 0.80}$      | $30.16 ^{\pm 0.26}$       |
> > > > > | $\ell_2$                         | $0.20$     | $2.33 ^{\pm 0.11}$  | $5.39 ^{\pm 0.34}$  | ${6.62 }^{\pm 0.07}$  | ${6.63 }^{\pm 0.12}$  | $6.10 ^{\pm 0.18}$          |
> > > > > | $\ell_2$                         | $0.50$     | $0.01 ^{\pm 0.01}$  | $0.06 ^{\pm 0.02}$  | ${0.07 }^{\pm 0.01}$         | ${0.10 }^{\pm 0.01}$  | $0.07 ^{\pm 0.02}$          |
> > > > > | $\ell_\infty$                    | $1/255$     | $10.29 ^{\pm 0.04}$ | $17.96 ^{\pm 1.08}$ | ${19.56 }^{\pm 0.33}$ | ${20.69 }^{\pm 0.81}$ | $18.63 ^{\pm 0.30}$         |
> > > > > | $\ell_\infty$                    | $2/255$     | $0.67 ^{\pm 0.01}$  | $1.96 ^{\pm 0.17}$  | ${2.16 }^{\pm 0.07}$  | ${2.62 }^{\pm 0.01}$  | $2.05 ^{\pm 0.17}$          |
> > > > > | $\ell_\infty$                    | $4/255$     | $0.01 ^{\pm 0.01}$  | $0.05 ^{\pm 0.01}$  | ${0.05 }^{\pm 0.00}$         | ${0.10 }^{\pm 0.01}$  | $0.05 ^{\pm 0.01}$          |
> > > > > | Clean acc.                       |            | $80.6^{\pm0.2}$     | $83.0^{\pm0.3}$     | ${84.0}^{\pm0.2}$     | $83.3^{\pm0.2}$              | ${83.9}^{\pm0.2}$    |

---

> > > > > > ### Comment · Reviewer_dJUQ · 2023-08-18
> > > > > >
> > > > > > Dear Authors,
> > > > > >
> > > > > > Thanks for your efforts. I'm glad to these encouraging results and have further updated my score.
> > > > > >
> > > > > > Best,

---

> > ### Comment · Reviewer_dJUQ · 2023-08-21
> > **Copy:  Official Comment by Reviewer dJUQ**
> >
> > Dear Authors,
> >
> > Thanks for the rebuttal. I believe that most of my concerns have been addressed, and I would like to raise my score when review editing is allowed.
> >
> > Here are some further questions for curious.
> >
> > In the context of adversarial robustness, a recent work aims to understand SAM as implicit adversarial training and revealed that SAM is beneficial for improving robustness [1]. On the other hand, there are several works attributing adversarial vulnerabilities of DNNs through normalization layers, and designed normalization layer-based adversarial training methods to improve adversarial robustness [2,3].
> >
> > My questions are:
> >
> > 1. Can SAM-ON and the corresponding observations in this paper be the connection between normalization layers and adversarial robustness?
> > 2. Can using SAM-ON only improve adversarial robustness?
> >
> > Feel free to dismiss these questions if you are not familiar with adversarial robustness. However, I believe discussing these questions in this paper can help you strengthen the understanding of the effectiveness of SAM-ON.
> >
> > [1] Sharpness-Aware Minimization Alone can Improve Adversarial Robustness. ICML 2023 workshop
> >
> > [2] Intriguing properties of adversarial training at scale. ICLR 2020
> >
> > [3] Batch Normalization Increases Adversarial Vulnerability and Decreases Adversarial Transferability: A Non-Robust Feature Perspective. ICCV 2021
> >
> > ---
> >
> > Dear authors, please paste the response below this comment. Thanks

---

> > > ### Author Response · Authors · 2023-08-21
> > > **Copy: Official Comment by Authors**
> > >
> > > Dear Reviewer,
> > >
> > > Following your suggestion, we have performed a preliminary study on the adversarial robustness of SAM-all and SAM-ON (with a WRN28-10 on CIFAR100). We find that both SAM-all and SAM-ON can significantly improve over SGD-trained models with respect to adversarial robustness. Interestingly, SAM-ON is slightly more robust than SAM-all, whereas ASAM-elem.-$\ell_\infty$-all is slightly more robust than ASAM-elem.-$\ell_\infty$-ON, but the differences are small and often within the standard deviation (reported over 3 seeds). Overall, we find that in order to get SAM-like improvements for adversarial robustness (as shown in https://arxiv.org/pdf/2305.05392.pdf) it is enough to only perturb the normalization layers in SAM, illustrating again their special role as outlined in our paper. These results are preliminary, but we found them interesting enough to report them already. Thank you again for the suggestion!
> > >
> > >
> > > | threat model                     | $\epsilon$ | SGD                 | SAM-all             | SAM-ON                       | ASAM-el.-$l_\infty$-all      | ASAM-el.-$l_\infty$-ON      |
> > > |----------------------------------|------------|---------------------|---------------------|------------------------------|------------------------------|-----------------------------|
> > > | $\ell_2$                         | $0.10$     | $18.14 ^{\pm 0.11}$ | $28.14 ^{\pm 1.09}$ | $	{31.28 }^{\pm 0.50}$  | ${30.33 }^{\pm 0.80}$      | $30.16 ^{\pm 0.26}$       |
> > > | $\ell_2$                         | $0.20$     | $2.33 ^{\pm 0.11}$  | $5.39 ^{\pm 0.34}$  | ${6.62 }^{\pm 0.07}$  | ${6.63 }^{\pm 0.12}$  | $6.10 ^{\pm 0.18}$          |
> > > | $\ell_2$                         | $0.50$     | $0.01 ^{\pm 0.01}$  | $0.06 ^{\pm 0.02}$  | ${0.07 }^{\pm 0.01}$         | ${0.10 }^{\pm 0.01}$  | $0.07 ^{\pm 0.02}$          |
> > > | $\ell_\infty$                    | $1/255$     | $10.29 ^{\pm 0.04}$ | $17.96 ^{\pm 1.08}$ | ${19.56 }^{\pm 0.33}$ | ${20.69 }^{\pm 0.81}$ | $18.63 ^{\pm 0.30}$         |
> > > | $\ell_\infty$                    | $2/255$     | $0.67 ^{\pm 0.01}$  | $1.96 ^{\pm 0.17}$  | ${2.16 }^{\pm 0.07}$  | ${2.62 }^{\pm 0.01}$  | $2.05 ^{\pm 0.17}$          |
> > > | $\ell_\infty$                    | $4/255$     | $0.01 ^{\pm 0.01}$  | $0.05 ^{\pm 0.01}$  | ${0.05 }^{\pm 0.00}$         | ${0.10 }^{\pm 0.01}$  | $0.05 ^{\pm 0.01}$          |
> > > | Clean acc.                       |            | $80.6^{\pm0.2}$     | $83.0^{\pm0.3}$     | ${84.0}^{\pm0.2}$     | $83.3^{\pm0.2}$              | ${83.9}^{\pm0.2}$    |

---

### Official Review · Reviewer_wExQ · 2023-07-07

**Soundness:** 3 good
**Presentation:** 3 good
**Contribution:** 3 good
**Rating:** 7
**Confidence:** 4

**Summary:**

This paper relates the normalization layer with the Sharpness-Awareness Minimization (SAM). Surprisingly, this paper finds that in the perturbation stage, only perturbing the affine parameters of normalization layers in the networks leads to a better generalization performance. Later, the authors investigate the reason behind such surprising phenomenon, and find that SAM-ON(only perturbing the normalization layer) even increase the sharpness, which doubts the intuition that the effectiveness of SAM comes from sharpness minimization. Extensive experiments demonstrate the effectiveness of the methods this paper proposed.

**Strengths:**

1. This paper identifies an important yet interesting problem: Is Sharpness-Awareness minimization effective because of minimizing sharpness? This paper found that only perturbing the normalization layer in the perturbation stage of SAM leads to both superior generalization performance and higher sharpness. The interact relationship between sharpness and generalization are quite doubtful.
2. This paper investigates the SAM from the normalization layer and found the normalization might be the reason for the success of SAM, which is quite surprising.
3. SAM with only perturbing the normalization layer largely saves the computational cost than the original SAM which needs perturbing all the parameters.

**Weaknesses:**

1. These experiments should be conducted on more network architectures, including VGG-Net and etc.
2. The observation that SAM with only perturbing the normalization layer holds the higher sharpness than original SAM might be trivial. SAM are designed to reduce the sharpness, while a degraded version of SAM should hold a higher sharpness in principle.
3. In this paper, it is clear that there is deep relationship between normalization layer and generalization, and there are indeed some papers on the relationship between normalization layer and generalization. Therefore, a related discussion should be included.

**Questions:**

1. In fact, the authors don't explain why SAM with only perturbing normalization layer works better than original SAM. Therefore, I would like the authors to include some discussion or theoretical analysis on that.

**Limitations:**

Yes. This paper discusses the limitation of this paper, where they only focus on the vision data.

---

> ### Author Rebuttal · Authors · 2023-08-09
>
> Thank you for your review and helpful suggestions.
> We address your concerns below.
>
> - __“more network architectures, including VGG-Net”__:
>   Following your suggestion, we have provided results for more neural network architectures: DenseNet and various VGG-Nets in Table 1 in the rebuttal pdf. Across the different architectures and SAM variants we consistently observe that SAM-ON outperforms regular SAM.
>
>
> - __“observation that … [SAM-ON] .. holds the higher sharpness than original SAM might be trivial. SAM are designed to reduce the sharpness, while a degraded version of SAM should hold a higher sharpness in principle.”__:
>   We agree this intuition is sensible, however prior work by Mi et al., 2022 hypothesizes that sparse SAM approaches can actually lead to a flatter landscape than regular SAM. It was thus worth investigating if SAM-ON (with its targeted, high sparsity level approach) would still be able to reduce sharpness. We will extend the discussion in the paper.
>
>
> - __“there is deep relationship between normalization layer and generalization .. Therefore, a related discussion should be included.”__
>   We agree. Some discussion on the connection between normalization layers and their affine parameters with generalization can already be found in the related work section, but we are keen to extend this further. If you have any specific papers in mind please let us know and we would be happy to discuss them.
>
>
> - __“the authors don't explain why SAM with only perturbing normalization layer works better than original SAM.”__:
>   We believe it is an important question for future work to understand the mechanism behind the success of not only SAM-ON but also regular SAM as our work lends support to the idea that sharpness-reduction may not be the sole source of SAM’s success. The search for alternative explanations for SAM’s success is an active area of research, for example recent work (that appeared on the arXiv after the NeurIPS submission deadline and hence not included in our related work section) suggests that the use of SAM prunes a significant number of activations and leads to low-rank features (arXiv:2305.16292). Our work contributes to this search by illustrating the success of only applying SAM to the normalization layers. We provide additional experiments in Figure 1 in the rebuttal pdf that explore potential connections of SAM-ON with weight decay and dropout. We find that SAM-ON’s improvements are not due to its interaction with weight decay (Figure 1, bottom left), nor can its success be mimicked by applying dropout solely to the normalization layers (Figure 1, top). We will include this discussion in the revised paper.

---

> > ### Comment · Reviewer_wExQ · 2023-08-12
> > **Thank you for response**
> >
> > I have thoroughly read the response and appreciate the detailed response to my questions.
> > - I have noticed the additional experiments on VGG-nets and DenseNets and the experimental results are sound to me.
> > - I agree that it is worth investigating that SAM-ON could still be able to reduce sharpness.
> > - I agree that it is important to study the underlying relationship between SAM and BN and would like to any future work on that.
> >
> > Regarding the authors' response and other reviewers' comments, I have raised my rating.

---

> > > ### Author Response · Authors · 2023-08-16
> > >
> > > Dear reviewer,
> > >
> > > Thanks for reading our rebuttal and checking our new results. We are happy we could address your concerns.

---

### Official Review · Reviewer_Ftd6 · 2023-07-08

**Soundness:** 3 good
**Presentation:** 2 fair
**Contribution:** 2 fair
**Rating:** 4
**Confidence:** 4

**Summary:**

This paper tries to analyze the effects of different layers for sharpness-aware minimization (SAM). The authors find that normalization layer plays an important role in the improvements of SAM and only perturbing the normalization layer can obtain a comparable result. Therefore, this paper proposes SAM-ON and the experimental results illustrate that SAM-ON can obtain a great performance on many different tasks with different models.

**Strengths:**

Strengths:
1. This paper focus on an important problem, SAM is very important for improving generalization.
2. The experimental finding is very interesting and can efficiently reduce the computation.
3. The authors try to evaluate the performance of proposed method on various datasets and models.

**Weaknesses:**

Weakness:
1. Although the proposed method can improve the performance on CIFAR. But CIFAR is too simple for SAM and that is weak to illustrate the improvement of SAM-ON. However, the results about resnet on imagenet is too close for me and the improvement is minimal. I'm not sure whether you run the experiments multiple times with different seeds.
2. The experiments on ViT is also a little weak for me. For ViT, cifar is too simple and you should train ViT form scratch on imagenet. Since SAM can achieve significant improvement on ViT+ImageNet and maybe you need to analyze it to obtain more convincing results.
3. Although SAM-ON can reduce the computation, the training efficiency cannot be significantly improved.

**Questions:**

Question:
1. Do you try to run the experiments for multiple times with different seeds.
2. You say only perturbing normalization layers can obtain similar results with perturbing all parameters. My question is do you try to only perturb the parameters outside the normalization layer?

---

> ### Author Rebuttal · Authors · 2023-08-09
>
> Thank you for your review and helpful suggestions.
> We address your concerns below.
>
> - __ImageNet results__:
> We agree with your comments on the ImageNet results. As you suggested, we have run a ViT from scratch on ImageNet (with additional evaluations on ImageNet-R and ImageNet-sketch) with both the Lion and AdamW optimizers and find that SAM-ON consistently outperforms SAM-all in this setting (see Table 2 in attached rebuttal pdf), we will include this in the revised paper. Our ImageNet results at present are indeed for a single seed (all other experiments are with multiple seeds), as we prioritized showing results across different SAM variants. Following your comment, we have started runs with different seeds. Preliminary results show very little variability (standard deviations of 0.047% and 0.007% for SAM-all and SAM-ON with RN50 and elementwise $\ell_\infty$). We will ensure to include multiple seeds for all results in the revised paper. Further, we agree that the results for ImageNet and ResNet are somewhat less impressive than for ViT, but would like to highlight that other SAM variants such as SSAM, ESAM, GSAM, and FisherSAM also obtain only marginal (or no) improvements in this setting, so this is within expectations.
>
> - __“CIFAR is too simple for SAM”__:
>   We would like to draw the reviewer’s attention to e.g. Table 1 of our paper, where it can be seen that for CIFAR-100 regular SAM (denoted SAM-all) can improve by a large margin over the vanilla optimizer. For several model types, CIFAR-100 is far from being solved. Therefore, the consistent improvement of SAM-ON over SAM-all across different models (see Table 1 in rebuttal pdf for additional VGG-Nets and DenseNet) and SAM variants on CIFAR data is remarkable and we believe worth reporting. We hope this in combination with the novel ImageNet and ViT results addresses the reviewer’s concerns, but please do not hesitate to let us know if you believe further experiments are missing.
>
> - __“Although SAM-ON can reduce the computation, the training efficiency cannot be significantly improved”__:
>   In Table 5 we observe that SAM-ON reduces wall-clock time compared to SAM by about 17%. We agree this is not a large improvement, although it does improve upon the sparse SAM approach by Mi et al., 2022. Further computational gains ($\approx$ 30%) may be achieved for our approach by applying SAM to only _part of_ the normalization layers. Although this leads to some reduction in test accuracy, the approach still strongly outperforms the base optimizer (see Figure 1, bottom right in rebuttal pdf). We are happy to include this discussion in this paper, but should highlight here that the main aim of our work was to shed light on the remarkable success of SAM and to illustrate the important role of the normalization layers in obtaining its enhanced generalization performance and the connection with sharpness. Although the reduced computation obtained using SAM-ON is a positive side-effect, it was not the main aim of our work and hence was not listed as one of our main contributions on p2.
>
> - __“My question is do you try to only perturb the parameters outside the normalization layer?”__:
>   Yes, this is denoted as no-norm (dotted lines) in Figure 1, 2, 6, and 7. no-norm typically performs similar to SAM-all, except for some perturbation variants like elementwise-$\ell_2$, where we observed a drastic drop in test accuracy. We briefly discuss this in Section 4 and 5.3, but will extend this discussion in the revised paper.

---

> > ### Author Response · Authors · 2023-08-16
> >
> > Dear reviewer,
> >
> > Thanks again for your review and helpful comments.
> > Apart from the layerwise perturbation model, all runs with three random seeds for ResNet50 on ImageNet have finished (see table below). Following your suggestion, we will include those in the revised paper.
> > Further, we would like to draw your attention to our previously posted rebuttal, in which thanks to your suggestions we show:
> > - consistent gains with a ViT trained from scratch on ImageNet using both Lion and AdamW as base optimizers, including an evaluation on OOD datasets.
> > - a discussion on the significance of our results, especially given the new experiments on CIFAR and ImageNet data reported in the rebuttal pdf.
> > - that a trade-off for further computational gains at the cost of somewhat reduced accuracy gains can be achieved by only applying SAM to part of the normalization layers.
> > - that _"only perturb the parameters outside the normalization layer"_ is shown as no-norm results in our paper.
> >
> >
> > Given the approaching end of the author-reviewer discussion period, we wanted to make sure our rebuttal addressed your concerns. Are there further questions we could help you with?
> >
> >
> > | top-1 | SGD                | SAM                | ESAM   | GSAM   | elem. $l_{2}$ | elem. $l_{2}$ | elem. $l_{\infty}$ | elem. $l_{\infty}$ |
> > |-------|--------------------|--------------------|--------|--------|--------------|--------------|--------------------|--------------------|
> > |       |                    | all                | all    | all    | all          | ON           | all                | ON                 |
> > | ResNet-50 | $77.03^{\pm0.13}$ | $77.65^{\pm0.11}$ | $77.05$ | ${77.20}$ | $77.65^{\pm 0.05}$ |  $\textbf{77.82}^{\pm 0.14}$ | $77.45^{\pm0.04}$ | $\textbf{77.82}^{\pm0.01}$ |

---

> > ### Comment · Reviewer_Ftd6 · 2023-08-20
> > **Thanks for your effort to make this paper more clear!**
> >
> > Thanks for your response and effort to make this paper more clear!
> >
> > I still have several concerns:
> >
> > 1. ImageNet results: I find the performance gain about imagenet training from scratch in your attached pdf is a little limited. For example: top 1 ACC: 71.33 -> 71.41. In my past experience about SAM, the improvement is usually significant for ViT training, such as gsam. So I think maybe the proposed method is not vet suitable for vit training.
> >
> > In addition, I find the most experiments are about ResNet, such as table 1,2,3. However, the accuracy on imagent is very close (SAM vs SAM-ON), although the improvement on cifar is clear. So I think maybe the proposed method can work well on simple tasks, but it is still not very clear for more complex tasks.
> >
> > Finally, I think big model may be easier to converge to sharp minima and we need to focus more on these big models and datasets. CIFAR is not enough for me.
> >
> > 2.  Only perturb the parameters outside the normalization layer: Thanks for your clarification. As you mentioned, no-norm typically performs similarly to SAM-all. Based on that, whether I can say the parameters outside the normalization layer provide the most properties that SAM can improve the performance? So that means the parameters outside the normalization layer are all you need and the normalization layer has some special properties. If that is true, maybe you need to provide more analysis.
> >
> > Overall, I think these paper mainly focus on the experimental analysis about different layers for SAM. So I think the author need to provide more results and analysis to make us convincing.

---

> > > ### Author Response · Authors · 2023-08-20
> > >
> > > Dear reviewer,
> > >
> > > Thank you for your response, we will address your concerns below.
> > >
> > > - Regarding your overall comment _“I think the author need to provide more results and analysis to make us convincing.”_:
> > >
> > >     Following the suggestions of all reviewers, we provided the following experiments in the rebuttal:
> > >     1. ViTs trained from scratch and evaluated on ImageNet and OOD datasets.
> > >     2. Multiple seeds for ResNet results on ImageNet.
> > >     3. Further computational speed-up can be achieved by only perturbing part of the normalization layers.
> > >     4. More models on CIFAR (3 VGG variants + DenseNet).
> > >     5. Evaluation of more sharpness measures.
> > >     6. SAM-ON achieves similar adversarial robustness as SAM (table provided in a comment to reviewer dJUQ).
> > >     7. Interaction with weight decay.
> > >     8. SAM-ON’s success can not be mimicked by applying dropout solely to the normalization layers.
> > >
> > >      In your original review you asked for specific experiments and clarifications (ImageNet from scratch with ViT, training efficiency, multiple seeds, perturb all but the normalization layers), which we all provided in our rebuttal (bulletpoint 1-3 above + explanations on no-norm).  In addition to the analysis in the main paper (Section 5, we investigate sparsity, sharpness, the effect on the affine parameter values and different training stages), we explored potential connections of SAM-ON with weight decay and dropout and its robustness to adversarial perturbations after insightful remarks by reviewer wExQ. We would further like to stress that the mechanism behind vanilla SAM’s success is still not well understood and that our work lends support to the idea that sharpness-reduction may not be the sole source of SAM’s success, which fits well into recent findings (https://arxiv.org/pdf/2302.07011.pdf and https://arxiv.org/abs/2305.16292) and hence contributes to an improved understanding of the method.
> > >
> > >     If there are any other specific experiments you would like to see, please let us know and we will gladly provide these in the revised paper.
> > >
> > > Regarding your specific comments:
> > >
> > > - _”most experiments are about ResNet, such as table 1,2,3.”_:
> > > Table 3 shows ViTs, and the gains are often even clearer than for ResNets. Further ViT results can be found in Table 4 (for fine tuning on ImageNet) and the rebuttal pdf (for ImageNet training from scratch + OOD evaluations).
> > >
> > >
> > > - Regarding 1. (ImageNet results):
> > > Our ImageNet results (for both ResNet and ViT) conclusively show that perturbing only the normalization layers (<0.1% of the total parameters) can outperform or at least perform on par compared to perturbing all the parameters. This is highly surprising and we believe an important finding that should be shared with the community as it provides new insights into SAM, given that the original motivation behind SAM is a sharpness measure defined with respect to _all_ parameters, and minimizing the respective measure only for an extremely small fraction of the total parameters would intuitively not lead to the same or even improved generalization benefits. There is already a significant improvement between SAM and the vanilla optimizer, and hence we think that any improvement of SAM-ON over this is fairly remarkable. Further gains can be found by changing the base SAM variant used (in Tables 1 and 3 we show that SAM-ON works for a range of different SAM-variants) and we would be happy to include results with GSAM in the revised paper (unfortunately this won’t be possible by the end of the discussion period, which is tomorrow).
> > > - Regarding 2. (no-norm results)
> > > Thank you for your new comment on the no-norm results. We believe it’s an important ablation study, which is why we included it in our paper. However, it is somewhat within expectations that perturbing all but the normalization layers, i.e., >99.9% of the parameters, would often give similar performance as SAM, given the findings of Mi et al., 2022. On the contrary, what is surprising is that perturbing only the normalization layers gives similar and in almost all cases better performance than SAM. We would also like to draw your attention to the fact that omitting the normalization layers _can_ lead to drastic performance decreases, as we show in our paper for ASAM elementwise-$\ell_2$.

---

### Author Rebuttal · Authors · 2023-08-09

We would like to thank you all for your time and useful comments. We are encouraged that you found that our paper addresses an interesting and important problem by demonstrating _"a new understanding of the underlying mechanism of SAM, which is both novel and informative”_ with _"extensive experiments [that] demonstrate the effectiveness of the methods"_.

Following your suggestions, in the pdf below we attach results for _“ViT [trained] from scratch on imagenet”_ with our SAM-ON method and also evaluate _“out of distribution robustness”_ (Table 2). For all datasets (ImageNet, ImageNet-R, ImageNet-sketch) and both Lion and AdamW optimizers, we observe that SAM-ON outperforms SAM-all. We also include results _“on more network architectures, including VGG-Net”_ (Table 1) and DenseNet on CIFAR data and again observe SAM-ON’s superiority across different SAM variants.

We address your other comments below and look forward to engaging with you in the discussion period.

---

### Author Response · Authors · 2023-08-21
**Summary of new experiments**

Dear all,

We would like to thank you all for your time and helpful remarks and suggestions. In order to facilitate your decision, we would like to summarize how we addressed your comments from both the reviews and the discussion phase:
1. We trained ViTs from scratch on ImageNet with both the AdamW and Lion optimizer and found consistent gains of SAM-ON over SAM-all and the vanilla optimizer (Table 2 in rebuttal pdf). To the best of our knowledge, we are the first to investigate the Lion+SAM combination. (Ftd6 asked for _”ViT form scratch on imagenet”_)
2. We additionally evaluated those models on the OOD datasets ImageNet-R and ImageNet-C, where again SAM-ON showed the best performance (reviewer sERj asked for  _”out-of-distribution robustness”_)
3. We trained a range of VGG architectures and a DenseNet on CIFAR10 and CIFAR100 (Table 1 in rebuttal pdf), where SAM-ON outperformed SAM and the vanilla optimizer consistently for all investigated SAM-variants. (reviewer wExQ asked for more _“more network architectures, including VGG-Net”_ )
4. We provided results with multiple seeds for the ResNet on ImageNet numbers. For ViTs, the runs haven’t finished yet, and we will provide them in the revised version. (reviewer Ftd6 asked for _”multiple seeds [for ImageNet]”_, Table is reported in response to their comment)
5. We report a range of other sharpness metrics in Table 3 (rebuttal pdf) and find in alignment with our previous findings that SAM-ON is sharper than SAM-all with respect to most metrics (especially worst-case sharpness and the metrics that are optimized during SAM), although there exist some exceptions. (reviewer sERj asked for _”other metrics of flatness”_)
6. We show that further computational gains may be achieved for our approach by applying SAM to only part of the normalization layers, cutting SAM’s additional cost roughly in half. Although this leads to some reduction in test accuracy, the approach still strongly outperforms the base optimizer (see Figure 1, bottom right in rebuttal pdf, half of additional cost corresponds to layer3 in purple). (reviewer Ftd6 commented on the training efficiency)
7. We find that in order to get SAM-like improvements for adversarial robustness (as shown in arxiv:2305.05392) it is enough to only perturb the normalization layers in SAM, illustrating again their special role as outlined in our paper. (reviewer dJUQ pointed us at a possible connection to adversarial robustness, Table is shown in response to their comment)
8. We provide experiments in Figure 1 in the rebuttal pdf that explore potential connections of SAM-ON with weight decay and dropout. We find that SAM-ON’s improvements are not due to its interaction with weight decay (Figure 1, bottom left), nor can its success be mimicked by applying dropout solely to the normalization layers (Figure 1, top). (Reviewer Ftd6 and wExQ asked for additional analysis)

Overall, we believe that SAM-ONs success is highly surprising and an important finding that should be shared with the community as it provides new insights into SAM: The original motivation behind SAM is a sharpness measure defined with respect to all parameters, and minimizing the respective measure only for an extremely small fraction of the total parameters should intuitively not lead to the same or even improved generalization benefits. We would further like to stress that the mechanism behind vanilla SAM’s success is still not well understood and that our work lends support to the idea that sharpness-reduction may not be the sole source of it, which fits well into recent findings (arXiv:2302.07011 and arxiv:2206.06232). Our work hence contributes to a search for alternative explanations by illustrating the success of only applying SAM to the normalization layers.

Thanks again for all your time and useful suggestions! We believe the new experiments provided in the rebuttal have made the paper even stronger, and we hope you will take this into consideration when making the final decision.
The authors

---

> ### Comment · Reviewer_dJUQ · 2023-08-21
> **Thanks for your efforts!**
>
> Dear authors,
>
> Thank you for your efforts during the rebuttal and discussion period. I really appreciate them and am still in favor in acceptance.
>
> I noticed that I unintentionally excluded other reviewers in the readers in our discussion. Sorry about that. I'll copy the discussion to make them visible to other reviewers.
>
> Best,

---

> > ### Author Response · Authors · 2023-08-21
> >
> > Dear reviewer,
> >
> > thanks for noticing! We have posted a copy of the comment with the results about the adversarial robustness experiment below your new comment.

---

### Decision · Program_Chairs · 2023-09-21

**Decision:**

Accept (poster)

**Comment:**

This paper discover an interesting phenomenon related to SAM optimizer which can highly improve its computational efficiency with comparable generalization performance or sometimes even better generalization than the original SAM algorithm. SAM is known to have a strong effect on generalization performance of deep neural networks, but it is computationally more expensive than SGD because in each iteration, SAM needs to compute the gradient twice. The key observation of this paper is that, instead of computing the gradient (that corresponds to the adversarial perturbation of the weights), one can perturb the affine parameters of batch normalization only, which can be around 0.1% of the number of weights in the models the paper studies. This is a huge reduction. Interestingly, the authors notice that other ideas that rely on a smaller number of parameters, such as sparse SAM, often lead to lower generalization performance compared to the proposed idea.

The ratings for the submission had a wide range, form 9 (Very Strong Accept) to 3 (Reject). I read all the reviews, specially the criticism raised by the reviewer sERj that gave the lowest rating. I also discussed these reviews with the Senior AC handling this submission. We eventually decided to accept the paper. The criticisms from  reviewer sERj were around lack of theory for the observed phenomena. However, this paper is an empirical paper, which discovers an interesting phenomenon and then backs it up with evidence spanning multiple datasets and different learning scenarios. As such, it is not necessary to have theory to get the paper accepted. This point was also mentioned as a reply to reviewer sERj by reviewer dJUQ. Based on this, and the fact that the submission has also received rating of 9 and 7, we recommend accept.